# Ensembles for Uncertainty Estimation: Benefits of Prior Functions and Bootstrapping

**Vikranth Dwaracherla**\*, **Zheng Wen**\*, **Ian Osband, Xiuyuan Lu,**
**Seyed Mohammad Asghari, Benjamin Van Roy**
*Efficient Agent Team, DeepMind, Mountain View, CA*

**Reviewed on OpenReview:** *https://openreview.net/forum?id=IqJsyulDUX*

## Abstract

In machine learning, an agent needs to estimate uncertainty to efficiently explore and adapt and to make effective decisions. A common approach to uncertainty estimation maintains an ensemble of models. In recent years, several approaches have been proposed for training ensembles, and conflicting views prevail with regards to the importance of various ingredients of these approaches. In this paper, we aim to address the benefits of two ingredients – prior functions and bootstrapping – which have come into question. We show that prior functions can significantly improve an ensemble agent's joint predictions across inputs and that bootstrapping affords additional benefits if the signal-to-noise ratio varies across inputs. Our claims are justified by both theoretical and experimental results.

## 1 Introduction

Effective decision making, exploration, and adaption often require an agent to know what it knows and also what it does not know, which in turn relies on the agent's capability to estimate uncertainty. For instance, in reinforcement learning (Sutton, 1988) and bandit (Lattimore & Szepesvári, 2020) problems, efficient exploration schemes such as upper-confidence bound (Lattimore & Szepesvári, 2020) and Thompson sampling (Russo et al., 2018) heavily rely on the agent's ability to model uncertainty. In the past few decades, many different approaches have been developed for uncertainty estimation, such as dropout (Gal & Ghahramani, 2016), Bayes by Backprop (Blundell et al., 2015), hypermodels (Dwaracherla et al., 2020), epinets (Osband et al., 2021), and stochastic Langevin MCMC (Welling & Teh, 2011; Dwaracherla & Van Roy, 2020). Maintaining an ensemble of models is one of the simplest and most widely used uncertainty estimation method. In particular, several variants of ensemble agents have been developed for uncertainty estimation (Lakshminarayanan et al., 2017; Fort et al., 2019; Osband et al., 2018) and downstream decision problems, such as bandits (Lu & Van Roy, 2017) and reinforcement learning (Osband et al., 2016). Compared to other uncertainty estimation approaches, the simplicity of ensemble agents makes it easier to analyze and implement in many practical problems. Moreover, since most uncertainty modeling methods face similar challenges, we believe that many insights gained from studying ensemble agents also translate to other methods, such as hypermodels and epinets.

An ensemble agent represents uncertainty based on a set of different models, but how should we train such different models based on the same training dataset and prior knowledge? Different approaches have been proposed, and it is still highly debated which approaches should be adopted for various practical problems. Some recent papers (Lakshminarayanan et al., 2017; Fort et al., 2019) suggest that we can train such models just with random parameter initialization and data shuffling. However, this approach does not pass some simple sanity checks; for example, it does not produce diversity across models when applied to linear regression. On the other hand, Osband et al. (2018) and He et al. (2020) propose to train ensemble agents with randomized *prior functions* and emphasize the importance of incorporating agent's prior uncertainty into

---

\*equal contribution

ensemble agent training. Finally, bootstrapping (Efron & Tibshirani, 1994) is a class of widely used techniques to train diverse models. However, a recent paper (Nixon et al., 2020) questions whether bootstrapping plays an essential role and suggests that it can even hurt performance. More broadly, the literature presents conflicting messages with regards to the importance of prior functions and bootstrapping. In this paper, we aim to bring clarity to these issues.

Most work on supervised learning has focused on producing accurate marginal predictions for each input. However, recent papers (Wen et al., 2021; Wang et al., 2021; Osband et al., 2022a) emphasize the importance of accurate joint predictions. In particular, accurate joint predictions are crucial for efficient exploration, adaptation and effective decision making. Taking cue from this line of research, we evaluate performance of ensemble agents based on their joint as well as marginal predictions.

We frame a class of ensemble agents – which includes those with and without prior functions or bootstrapping – through proposing *loss function perturbation* as a unifying concept. By systematically comparing performance of agents across this class, we elucidate benefits of prior functions and bootstrapping. In particular, we show that prior functions can significantly enhance an ensemble agent's joint predictions. As explained in (Wen et al., 2021), joint predictions drive efficiency of exploration and adaptation. With regards to bootstrapping, we show that further benefits depend on the variation of the signal-to-noise ratio (SNR) across the input space. If the SNR is uniform across the input space and prior functions are appropriately tuned then bootstrapping offers no significant value. On the other hand, if the SNR varies substantially across the input space then bootstrapping can significantly improve performance regardless of how prior functions are tuned.

The remainder of this paper proceeds as follows: we first review literature in Section 2 and formulate the problem in Section 3. Then we justify the main points of this paper, as discussed above, via analysis on Bayesian linear regression (Section 4), numerical experiments on classification using neural networks (Section 5) and bandit problems (Section 6). For experiments on classification we consider the synthetic data generated by Neural Testbed (Osband et al., 2022a), and CIFAR10 (Krizhevsky, 2009), a widely used benchmark image dataset. Finally, Section 7 concludes the paper.

## 2  Literature review

In recent years, in addition to improving prediction accuracy (Hastie et al., 2009; Dietterich, 2000), ensemble agents have also been built for uncertainty estimation (Lakshminarayanan et al., 2017; Fort et al., 2019; Osband et al., 2018) and decision-making (Lu & Van Roy, 2017). In particular, Lakshminarayanan et al. (2017) proposed to train ensemble agents for uncertainty estimation only based on random parameter initialization and data shuffling. Though this simple approach might work in certain scenarios, it does not pass simple sanity checks as shown in Section 4. On the other hand, Osband et al. (2018); He et al. (2020) proposed to train ensemble agents by effectively perturbing the loss function of each model with a random but fixed additive *prior function*, which incorporates the agent's prior uncertainty.

There is a long line of research on the topic of diversity in deep ensembles. Specifically, Shui et al. (2018) proposes to train deep ensembles with a diversity function regularization. Jain et al. (2020) proposes to train ensemble models for uncertainty estimation by maximizing "overall diversity". Nam et al. (2021) studies knowledge distillation (KD) for deep ensembles, and proposes a new approach based on Output Diversified Sampling (ODS). D'Angelo & Fortuin (2021) studies repulsive deep ensembles, which added a kernelized repulsive term in the update rule of the deep ensembles. It also shows that the training dynamics of the proposed approach follow a Wasserstein gradient flow of the KL divergence to the true posterior. Rame & Cord (2021) introduces a novel training criterion called DICE, which increases the diversity by reducing spurious correlations among features. Theoretical properties of ensembles with prior functions have previously been discussed in Ciosek et al. (2020). However, we believe that theoretical insights there suffer from some major limitations: first, Ciosek et al. (2020) assumes that uncertainty is independent across different inputs. Ideally if we expect the uncertainty at one input is high, so should the uncertainty at another input that is close. However, this is not captured by the method proposed in Ciosek et al. (2020). Second, Ciosek et al. (2020) assumes that uncertainty only depends on the input. Though this assumption is reasonable for the linear setting, it is quite restrictive for more complex models.

Ensemble agents are also widely used in sequential decision problems, such as bandits (Lu & Van Roy, 2017) and reinforcement learning (Osband et al., 2016). In particular, ensemble sampling (Lu & Van Roy, 2017; Qin et al., 2022) is a particular version of Thompson sampling (Thompson, 1933; Russo et al., 2018) that uses an ensemble of models to represent the uncertainty.

In most existing literature, agents are evaluated based on their *marginal* predictive distributions. Examples of such metrics include marginal negative log-likelihood (NLL), Brier score (Brier et al., 1950), accuracy, and expected calibration error (ECE) (Guo et al., 2017). Recent papers (Wen et al., 2021; Wang et al., 2021) discussed the importance of joint predictive distribution for a wide range of downstream tasks, such as sequential decision problems. Practical evaluation metrics have also been developed to evaluate the joint predictive distributions (Osband et al., 2022a;b). A recent paper (Nixon et al., 2020) suggests that non-parametric bootstrapping might not be helpful for building better ensemble agents. However, this paper is limited to small ensemble sizes, high-SNR deep learning problems, and the agents are evaluated based on their marginal predictive distributions. In addition to ensemble agents, there are other agents developed for uncertainty estimation, as discussed in Section 1. Recently, the concept of epistemic neural network (Osband et al., 2021) has been developed as a unifying umbrella for these agents.

## 3   Preliminaries

Consider a sequence of data pairs $((X_t, Y_{t+1}) : t = 0, 1, 2, \ldots)$, where each $X_t$ is an input vector and $Y_{t+1}$ is the associated output. Each output $Y_{t+1}$ is independent of all other data, conditioned on $X_t$, and distributed according to $\mathcal{E}(\cdot|X_t)$. The conditional distribution $\mathcal{E}$ is referred to as the *environment*. The environment $\mathcal{E}$ is random; and this reflects the agent's uncertainty about how outputs are generated given inputs.

Note that $\mathbb{P}(Y_{t+1} \in \cdot|\mathcal{E}, X_t) = \mathcal{E}(\cdot|X_t)$ and $\mathbb{P}(Y_{t+1} \in \cdot|X_t) = \mathbb{E}[\mathcal{E}(\cdot|X_t)|X_t]$.

| agent | random initialization | prior function | bootstrapping |
|---|:---:|:---:|:---:|
| ensemble-N | Yes | No | No |
| ensemble-P | Yes | Yes | No |
| ensemble-BP | Yes | Yes | Yes |

Table 1: Comparison of ensemble-N, ensemble-P, and ensemble-BP: whether or not random initialization, prior function, and bootstrapping are used for training different models.

To illustrate the main points of this paper, we consider three variants of ensemble agents, referred to as ensemble-N, ensemble-P, and ensemble-BP[1]. All these agents include an ensemble of models, and are trained based on $\mathcal{D}_T \equiv ((X_t, Y_{t+1}) : t = 0, 1, \ldots, T - 1)$. However, they differ in if prior functions and bootstrapping are used for training, as illustrated in Table 1. In particular, the models in ensemble-N are trained with the same loss function but different random initialization. On the other hand, in addition to random initialization, models in ensemble-P are trained with random prior functions, and models in ensemble-BP are trained with both prior functions and bootstrapping. Note that we will not consider an ensemble agent that is trained with bootstrapping alone and without random prior functions. This is because when there is no data (e.g. in cold-start problems), bootstrapping alone cannot induce any uncertainty, irrespective of the bootstrapping noise level.

For each ensemble agent, we denote its $m$-th model as $\hat{\mathcal{E}}_m$, and hence the agent is represented by $\Theta_T = (\hat{\mathcal{E}}_1, \ldots, \hat{\mathcal{E}}_M)$, where $M$ is the number of models in the ensemble. Note that an ensemble agent defines a discrete distribution in the space of environments:

$$\mathbb{P}(\hat{\mathcal{E}} \in \cdot|\Theta_T) = \tfrac{1}{M} \sum_{m=1}^{M} \mathbf{1}(\hat{\mathcal{E}}_m \in \cdot),$$

where $\hat{\mathcal{E}}$ is the agent's model of the environment. This discrete distribution is an approximation of the true posterior $\mathbb{P}(\mathcal{E} \in \cdot|\mathcal{D}_T)$, and characterizes the agent's uncertainty about $\mathcal{E}$.

---

[1]"P" stands for prior function, "B" for bootstrapping, and "N" for neither.

We now briefly discuss how to evaluate an ensemble agent's performance. In supervised learning, one natural metric is the expected Kullback–Leibler (KL) divergence between $\mathbb{P}(\mathcal{E} \in \cdot | \mathcal{D}_T)$ and $\mathbb{P}(\hat{\mathcal{E}} \in \cdot | \Theta_T)$. In Section 4, we use this metric to evaluate ensemble agents with a large number of models in linear regression problems. However, for more general problems, this metric is computationally intractable and can also be infinite. Thus, following previous papers (Wen et al., 2021; Osband et al., 2022a;b), we evaluate the ensemble agent based on its predictive distributions. Specifically, for any inputs $X_{T:T+\tau-1} \equiv (X_T, \ldots, X_{T+\tau-1})$, the ensemble agent determines a predictive distribution, which could be used to sample imagined outcomes $\hat{Y}_{T+1:T+\tau} \equiv (\hat{Y}_{T+1}, \ldots, \hat{Y}_{T+\tau})$. Hence, the agent's $\tau$-th order predictive distribution is given by

$$\hat{P}_{T:T+\tau-1} = \mathbb{P}(\hat{Y}_{T+1:T+\tau} \in \cdot | \Theta_T, X_{T:T+\tau-1}), \tag{1}$$

which represents an approximation to the underlying output distribution that would be obtained by conditioning on the environment:

$$P^*_{T:T+\tau-1} = \mathbb{P}(Y_{T+1:T+\tau} \in \cdot | \mathcal{E}, X_{T:T+\tau-1}). \tag{2}$$

If $\tau = 1$, this represents a marginal prediction; that is a prediction of a single output. For $\tau > 1$, this is a joint prediction over a sequence of $\tau$ outputs. Recent papers (Wen et al., 2021; Osband et al., 2022a) propose to use the expected KL-divergence between $P^*_{T:T+\tau-1}$ and $\hat{P}_{T:T+\tau-1}$, referred to as $\mathbf{d}^\tau_{\mathrm{KL}}$, to evaluate both the marginal prediction and the joint prediction. Osband et al. (2022b) further proposed a variant of $\mathbf{d}^\tau_{\mathrm{KL}}$ based on *dyadic sampling*, referred to as $\mathbf{d}^{\tau,2}_{\mathrm{KL}}$, to evaluate the agents' joint prediction for practical problems with high input dimension. In Section 5, we follow Osband et al. (2022b) to use $\mathbf{d}^\tau_{\mathrm{KL}}$ with $\tau = 1$ to evaluate marginal predictions and $\mathbf{d}^{\tau,2}_{\mathrm{KL}}$ with $\tau = 10$ to evaluate joint predictions. Finally, in bandit problems (Section 6), we use the standard expected cumulative regret to evaluate ensemble agents. We will further discuss the evaluation metrics in subsequent sections and Appendix A.

## 4 Bayesian linear regression

In this section, we use the classical Bayesian linear regression problem as a didactic example to compare `ensemble-N`, `ensemble-P`, and `ensemble-BP` with a large number of models. In particular, we highlight the benefits of bootstrapping in `ensemble-BP`.

Consider a Bayesian linear regression problem with heteroscedastic noises:

$$Y_{t+1} = \theta_*^T X_t + W_{t+1}, \quad t = 0, 1, 2, \ldots \tag{3}$$

where $\theta_* \in \Re^d$ is the coefficient vector and identifies the environment $\mathcal{E}$, and $W_{t+1}$'s are additive noises. We assume that inputs $X_t$'s are i.i.d. sampled from an input distribution $P_X$. Moreover, conditioned on $X_t$, the additive noise $W_{t+1}$ is independently sampled from $N(0, \sigma^2(X_t))$, where $\sigma^2 : \Re^d \to \Re^+$ is the noise variance function, i.e., noise variance depends on input. The agent does not know $\theta_*$, but knows that prior over $\theta_*$ is $N(0, \sigma_0^2 I)$[2] and also knows $\sigma^2(\cdot)$. Conditioned on the training data $\mathcal{D}_T$, the posterior over $\theta_*$ is $\theta_* | \mathcal{D}_T \sim N(\mu_T, \Sigma_T)$, where the posterior mean $\mu_T$ and the posterior covariance $\Sigma_T$ are

$$\Sigma_T = \left[ I/\sigma_0^2 + \sum_{t=0}^{T-1} X_t X_t^T / \sigma^2(X_t) \right]^{-1} \quad \text{and} \quad \mu_T = \Sigma_T \left[ \sum_{t=0}^{T-1} X_t Y_{t+1} / \sigma^2(X_t) \right].$$

### 4.1 Ensemble agents with a large number of models

We consider ensemble agents with a large number of models, and each $m$-th model is identified by a coefficient vector $\hat{\theta}_m \in \Re^d$. As a standard method for training ensemble agents on linear regression problems (Lu & Van Roy, 2017; Qin et al., 2022), we compute $\hat{\theta}_m$ by minimizing the following perturbed loss function:

$$\hat{\theta}_m \in \arg\min_\theta \sum_{t=0}^{T-1} \frac{\nu(X_t)}{2} \left( \theta^T X_t - [Y_{t+1} + Z_{t+1,m}] \right)^2 + \frac{\lambda}{2} \left\| \theta - \tilde{\theta}_m \right\|_2^2, \quad \forall m = 1, \ldots, M \tag{4}$$

---

[2]Note that this isometric prior assumption is without loss of generality: any Bayesian linear regression problem satisfies this assumption after appropriate coordinate transformation.

where $Z_{t+1,m}$ is additive perturbation for data pair $(X_t, Y_{t+1})$ sampled from $N\left(0, \hat{\sigma}^2(X_t)\right)$; $\nu, \hat{\sigma}^2 : \Re^d \to \Re^+$ are respectively the weight function and the bootstrapping variance function; $\lambda \geq 0$ is the regularization coefficient; and $\tilde{\theta}_m$ is independently sampled from $N(0, \hat{\sigma}_0^2 I)$. The magnitude of $\hat{\sigma}_0^2$ reflects the agent's prior uncertainty, and hence $\hat{\sigma}_0$ is known as the agent's *prior scale*. Note that $\|\theta - \tilde{\theta}_m\|_2^2$ is a randomly perturbed regularizer, and is equivalent to a prior function for linear regression (Osband et al., 2018). It is straightforward to show that

$$\hat{\theta}_m = \left(\sum_{t=0}^{T-1} \nu(X_t) X_t X_t^T + \lambda I\right)^{-1} \left(\sum_{t=0}^{T-1} \nu(X_t)\left(Y_{t+1} + Z_{t+1,m}\right) X_t + \lambda \tilde{\theta}_m\right). \tag{5}$$

Note that $Z_{t+1,m}$ and $\tilde{\theta}_m$ are random variables, and $\hat{\theta}_m \,|\, \mathcal{D}_T \sim N\left(\hat{\mu}_T, \hat{\Sigma}_T\right)$, where $\hat{\mu}_T$ and $\hat{\Sigma}_T$ are respectively the mean and covariance of $\hat{\theta}_m$ conditioned on $\mathcal{D}_T$ (see Appendix B.1 for their analytical solutions). We say an ensemble agent is *unbiased* if $\hat{\mu}_T = \mu_T$.

As discussed in Section 1, we consider three variants of ensemble agents: `ensemble-N`, `ensemble-P`, and `ensemble-BP`. Note that for Bayesian linear regression, all these three agents can be trained based on the perturbed loss function (4), but with different constraints on loss function parameters reflecting if prior functions and bootstrapping are enabled. In particular, `ensemble-N` does not use prior functions or bootstrapping, and fix $\hat{\sigma}_0^2 = 0$ and $\hat{\sigma}^2(\cdot) = 0$. `ensemble-P` uses prior functions, but does not use bootstrapping by fixing $\hat{\sigma}^2(\cdot) = 0$. `ensemble-BP` uses both prior functions and bootstrapping. Table 2 summarizes the differences between `ensemble-N`, `ensemble-P`, and `ensemble-BP`.

| agent | $\lambda$ and $\nu(\cdot)$ | $\hat{\sigma}_0^2$ | $\hat{\sigma}^2(\cdot)$ |
|---|---|---|---|
| `ensemble-N` | allowed to tune | fixed at 0 | fixed at 0 |
| `ensemble-P` | allowed to tune | allowed to tune | fixed at 0 |
| `ensemble-BP` | Proposition 1 | Proposition 1 | Proposition 1 |

Table 2: Comparison of `ensemble-N`, `ensemble-P`, and `ensemble-BP` for linear regression. The parameters for `ensemble-BP` are chosen as the optimal parameters (Proposition 1).

Note that the discrete distribution represented by an ensemble agent is an *empirical distribution* of $M$ samples from $P\left(\hat{\theta}_m \in \cdot \,|\, \mathcal{D}_T\right) = N\left(\hat{\mu}_T, \hat{\Sigma}_T\right)$. Moreover, as $M \to \infty$, this discrete distribution converges to $P\left(\hat{\theta}_m \in \cdot \,|\, \mathcal{D}_T\right)$. In this section, we choose to evaluate

$$\mathbb{E}\left[\mathbf{d}_{\mathrm{KL}}\left(P\left(\theta_* \in \cdot \,|\, \mathcal{D}_T\right) \,\|\, P\left(\hat{\theta}_m \in \cdot \,|\, \mathcal{D}_T\right)\right)\right], \tag{6}$$

which effectively measures the performance of an ensemble agent with a large number of models. Further discussion on this metric is provided in Appendix A. The following results show that for Bayesian linear regression, `ensemble-N` cannot represent any uncertainty, since its models use the same strictly convex loss function. On the other hand, `ensemble-BP` with infinite models is able to match the exact posterior $\mathbb{P}\left(\theta_* \in \cdot \,|\, \mathcal{D}_T\right)$ by properly choosing the parameters in the perturbed loss function (4). The proofs are straightforward and are provided in Appendix B.2.

**Proposition 1** (`ensemble-BP`) *For perturbed loss function* (4), *if we choose* $\hat{\sigma}^2(\cdot) = \sigma^2(\cdot)$, $\nu(\cdot) = 1/\sigma^2(\cdot)$, $\hat{\sigma}_0^2 = \sigma_0^2$, *and* $\lambda = 1/\sigma_0^2$, *then* $\mathbb{P}\left(\hat{\theta}_m \in \cdot \,\big|\, \mathcal{D}_T\right) = \mathbb{P}\left(\theta_* \in \cdot \,|\, \mathcal{D}_T\right)$.

**Proposition 2** (`ensemble-N`) *For perturbed loss function* (4), *if we choose* $\hat{\sigma}^2(\cdot) = 0$ *and* $\hat{\sigma}_0^2 = 0$, *then* $\hat{\theta}_m$ *is deterministic conditioned on* $\mathcal{D}_T$.

## 4.2 Benefits of bootstrapping

We now compare `ensemble-P` and `ensemble-BP`, and show the benefits of bootstrapping. Note that Proposition 1 indicates that by appropriately choosing parameters for `ensemble-BP`, $\mathbb{P}\left(\hat{\theta}_m \in \cdot \,\big|\, \mathcal{D}_T\right) = \mathbb{P}\left(\theta_* \in \cdot \,|\, \mathcal{D}_T\right)$. On the other hand, we derive a lower bound on the expected KL-divergence for unbiased `ensemble-P`, based

on expected signal-to-noise (SNR) ratios along orthogonal directions. Consider the positive semi-definite matrix

$$\Gamma \equiv \mathbb{E}\left[\sigma_0^2 X_t X_t^T / \sigma^2(X_t)\right], \tag{7}$$

where the expectation is over the input $X_t$. Let $\gamma_i$, $i = 1, 2, \ldots, d$, denote the $d$ eigenvalues of $\Gamma$, and let $v_i$ denote the normalized eigenvector associated with $\gamma_i$. Recall that $\gamma_i$'s are nonnegative and $v_i$'s are orthonormal. Hence, we have $\theta_*^T X_t = \sum_{i=1}^d \left(\theta_*^T v_i\right)\left(X_t^T v_i\right)$ and

$$\gamma_i = v_i^T \Gamma v_i = \mathbb{E}\left[\frac{\sigma_0^2 (X_t^T v_i)^2}{\sigma^2(X_t)}\right] = \mathbb{E}\left[\frac{\mathbb{V}\left[(X_t^T v_i)(\theta_*^T v_i) \mid X_t\right]}{\sigma^2(X_t)}\right],$$

where $\mathbb{V}[\cdot]$ denotes the (conditional) variance. Note that $\mathbb{V}\left[(X_t^T v_i)(\theta_*^T v_i) \mid X_t\right]$ can be interpreted as the magnitude of signal at $X_t$ along direction $v_i$, and $\sigma^2(X_t)$ is the magnitude of noise at $X_t$. Thus, eigenvalue $\gamma_i$ can be interpreted as expected SNR along direction $v_i$. We have the following lower bound for unbiased `ensemble-P` based on $\gamma_i$'s:

**Theorem 1** *For perturbed loss function ([4](#)) and `ensemble-P` agents with $\hat{\sigma}^2(\cdot) = 0$, under any choice of $\nu(\cdot)$, $\lambda$, and $\hat{\sigma}_0^2$ such that `ensemble-P` is unbiased, we have*

$$\mathbb{E}\left[\mathbf{d}_{\mathrm{KL}}\left(P\left(\theta_* \in \cdot \middle| \mathcal{D}_T\right) \middle\| P\left(\hat{\theta}_m \in \cdot \middle| \mathcal{D}_T\right)\right)\right] \geq \frac{1}{2}\sum_{i=1}^d \ln\left(\frac{1 + T\bar{\gamma}}{1 + T\gamma_i}\right), \tag{8}$$

*where $\gamma_i$'s are $d$ eigenvalues of $\Gamma$ and $\bar{\gamma} = \frac{1}{d}\sum_{i=1}^d \gamma_i$.*

Please refer to Appendix B.3 for the proof of Theorem 1. Note that this lower bound is always non-negative due to Jensen's inequality, and is zero when $\gamma_i$'s are equal. Theorem 1 states that if expected SNRs along different directions are very different, or equivalently if matrix $\Gamma$ is "ill-conditioned", then the expected KL defined in (6) is large for unbiased `ensemble-P`. Note that $\Gamma$ can be ill-conditioned for several different reasons. For instance, $\Gamma$ is likely to be ill-conditioned if $\sigma^2(X_t)$ varies significantly across different directions (heteroscedastic), or if the covariance matrix of the input distribution $P_X$ is ill-conditioned.

Theorem 1 is a general lower bound and demonstrates the fundamental limitation of `ensemble-P` and potential benefits of bootstrapping. However, there are special cases where bootstrapping is not strictly necessary and `ensemble-P` can perform well after appropriate parameter tuning for each setting. Please refer to Appendix B.4 for such examples.

## 5 Classification on Synthetic and Real Data

In this section, we switch the gear to compare `ensemble-N`, `ensemble-P`, and `ensemble-BP` in classification problems. In specific, we look at classification problems on an open-source test suite called Neural Testbed (Osband et al., 2022a) and a widely used benchmark image dataset, CIFAR10 (Krizhevsky, 2009). More details about Neural Testbed and CIFAR10 dataset are provided in Appendices C and D. Our experiment results show that many insights developed in Section 4 also translate to these classification problems. It is worth mentioning that in this section, we consider ensemble agents with a limited number of models, and each model is a neural network.

In Section 4, we have seen that `ensemble-N` reduces to a single point estimate. This might be a bit contrived as the loss function is not convex in many deep learning problems. One would hope that, with neural networks, random initialization is sufficient and `ensemble-N` will perform well, as suggested by Lakshminarayanan et al. (2017). Indeed, in our experiments, we find that `ensemble-N` improves over the performance of a single point estimate. However, with prior functions and bootstrapping, the performance of ensemble agents can be further improved significantly.

Given a dataset $\mathcal{D}_T = (X_t, Y_{t+1})_{t=0}^{T-1}$, the $m$-th model in an ensemble agent for the classification problems aims to minimize the perturbed loss function of form

$$\mathcal{L}(\theta_m, \mathcal{D}_T) = -\sum_{t=0}^{T-1} W_{t+1,m} \log \left( \frac{\exp\left(f_{\theta_m}(X_t)\right)_{Y_{t+1}}}{\sum_{i=1}^{|\mathcal{Y}|} \exp\left(f_{\theta_m}(X_t)\right)_i} \right) + \lambda \|\theta_m\|_2^2, \tag{9}$$

where $W_{t+1,m}$ is the weight corresponding to the data pair $(X_t, Y_{t+1})$ and the $m$-th model in the ensemble, $f_\theta(X_t) \in \Re^{|\mathcal{Y}|}$ is the vector of logits for input $X_t$, and $\mathcal{Y}$ is the set of possible labels. Note that the weights $W_{t+1,m}$ are sampled i.i.d, for each model $m$ and data pair $(X_t, Y_{t+1})$, at the start and remain fixed for the entire experiment. For both `ensemble-N` and `ensemble-P` agents, $W_{t+1,m} = 1 \ \forall t, m$, while for an `ensemble-BP` agent, $W_{t+1,m}$ are sampled i.i.d. from $2 \times \text{Bernoulli}(0.5)$.

## 5.1 Dyadic Sampling

We evaluate the agents on both marginal predictions and joint predictions. In specific, we use dyadic sampling (Osband et al., 2022b) to evaluate agents on joint predictions. We provide a high-level overview of dyadic sampling here, and please refer to Appendix A for more details.

Dyadic sampling is a computationally efficient approach to evaluate joint predictions, which provides additional insights beyond marginal predictions. In a high level, dyadic sampling proceeds as follows: it first samples two random *anchor points* from the input space, and then resamples $\tau$ inputs from these two anchor points. The expected KL-divergence under dyadic sampling is referred to as $\mathbf{d}_{\text{KL}}^{\tau,2}$. Similar to Osband et al. (2022b), we use dyadic sampling with $\tau = 10$.

Moreover, as discussed in recent papers (Wen et al., 2021; Osband et al., 2022b), expected KL-divergence is equivalent to negative log-likelihood (nll), which is also known as the *cross-entropy loss*, for agent comparison. Thus, for experiments on the CIFAR10 dataset, where the likelihood under the true environment is unknown, we use cross-entropy loss under dyadic sampling to compare agents.

## 5.2 Experiments on Neural Testbed

We first compare `ensemble-N`, `ensemble-P`, and `ensemble-BP` on Neural Testbed to demonstrate the benefits of prior functions and bootstrapping in different scenarios. When comparing agents on Neural Testbed, the network for the single point estimate is chosen to have the same architecture as the generative model, a 2-layer MLP. We refer to this agent as `mlp` agent. The `ensemble-N` agent uses an ensemble of 2-layer MLPs. The `ensemble-P` agent uses an ensemble of single point estimates combined with additive prior function (Osband et al., 2018) at logits. The additive prior functions for each model of `ensemble-P` agent are different only through initialization. In particular, we use random MLPs, with the same architecture as the generative model, as the prior functions for experiments on Neural Testbed. The `ensemble-BP` agent uses the same model as an `ensemble-P` agent, but uses bootstrapping to train the model. Please find more details about Neural Testbed and agents in Appendix C.

Figure 1 compares `mlp`, `ensemble-N`, `ensemble-P`, and `ensemble-BP` agents on Neural Testbed problems with input dimension 100. All the results are normalized w.r.t performance of `mlp` agent. We consider two ways of tuning the hyperparameters, per problem and global tuning, where we pick a single hyperparameter set for all Neural Testbed problems. We observe that in both types of tuning, all the agents perform similarly in marginal predictions. None of the ensemble agents are statistically distinguishable from an `mlp` which uses a single point estimate. However, when evaluated by joint predictions, the agents are clearly distinguished. When we pick the best hyperparameter per problem (after averaging over 10 random seeds), we can see that `ensemble-N` offers some advantage over `mlp`. This is significantly improved by `ensemble-P` which uses prior functions. This highlights the benefits of prior functions. With bootstrapping, `ensemble-BP` performs very similarly to `ensemble-P` and doesn't improve the performance further. However, when a single hyperparameter is used over the entire testbed, `ensemble-BP` agent performs better than `mlp` and other ensemble agents. This indicates that one benefit of bootstrapping is robustness to parameter tuning.

As presented in Section 4.2, for classification problems with different SNRs across inputs, bootstrapping can further enhance the performance of ensemble agents. To illustrate this, we consider Neural Testbed problems

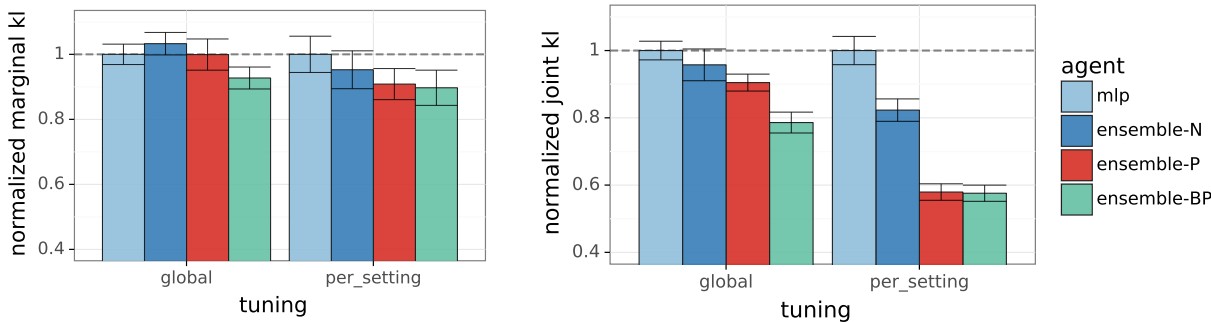

Figure 1: Marginal predictions do not distinguish agents. Prior functions significantly improve the joint predictions, with per setting tuning. Bootstrapping increases robustness to parameter specification and improves predictions, with global tuning.

with medium temperature 0.1, and with 25% of samples with label 1 in the training dataset randomly flipped to label 0 for each problem. This essentially generates two different SNRs across inputs for each problem. We average the results over 100 random seeds. Figure 2 shows the performance of `ensemble-P` and `ensemble-BP` agents on marginal and joint predictions. Best hyperparameters are chosen per setting for both the agents. We observe that `ensemble-BP` agent doesn't show a statistically significant improvement over `ensemble-P` in marginal predictions, but does show such an improvement in joint predictions. A more detail breakdown of the performance across different data regimes and results from additional experiments are provided in Appendix C.2.

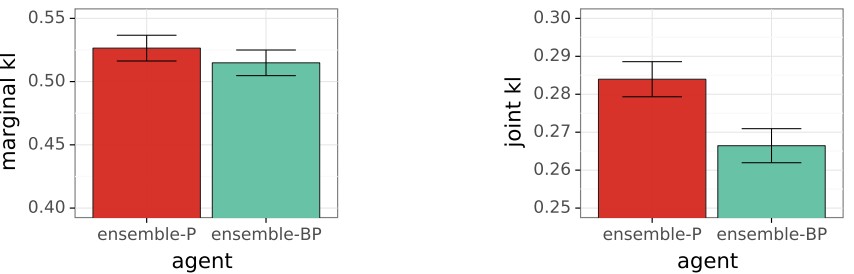

Figure 2: Marginal and joint KL on Neural Testbed problems with varying SNR.

## 5.3 Experiments on CIFAR10

In Section 5.2, we have seen results highlighting the benefits of prior functions and bootstrapping on Neural Testbed. Neural Testbed offers us a simple and transparent setting and helps us do systematic research and gain insights. However, at the end the ensemble agents need to be applied on practical problems. We compare the considered ensemble agents on CIFAR10 dataset and show the benefits of prior functions and bootstrapping.

CIFAR10 has been widely known in the deep learning community as a benchmark dataset and used to test classification algorithms. We look at the average performance across 4 problems, each with a different training dataset size from $\{10, 100, 1000, 50000\}$, where $50000$ is the size of full training data set. For these experiments, we use a `vgg` model (Simonyan & Zisserman, 2014) as the single point estimate model. An `ensemble-N` agent consists of an ensemble of `vgg` models and `ensemble-P` agent consists of an ensemble of models with each model being a `vgg` model combined with a small randomly initialized convolution network at logits. The `ensemble-BP` agent uses the same network architecture as `ensemble-P` agent, but uses loss functions perturbed via bootstrapping. In specific, we choose weights in Equation 9 as $W_{t+1,m} \sim \text{Bernoulli}(p)/p$. We consider three variants with $p = 0.5$, $p = 0.75$, and $p = 0.9$, and the respective agents are referred to

as `ensemble-BP`(0.5), `ensemble-BP`(0.75), and `ensemble-BP`(0.9). More details about the agents and the CIFAR10 dataset are provided in Appendix D.

Figure 3 compares the `vgg`, `ensemble-N` and `ensemble-BP` agents. The ensemble agents use 100 models, averaged across 5 random seeds, and we normalize performance w.r.t `vgg`. Similar to Figure 1, we observe that both `ensemble-N` and `ensemble-P` perform similarly on marginal predictions and offer some improvement over a single point estimate, `vgg`. However, once evaluated by joint predictions, `ensemble-P` performs much better than `ensemble-N`, and both `ensemble-N` and `ensemble-P` improve over `vgg`. The agents are evaluated by negative log-likelihood, which is equivalent to expected KL divergence for agent comparison (Appendix A.3). A detail breakdown of performance across different problems is provided in Appendix D.2.

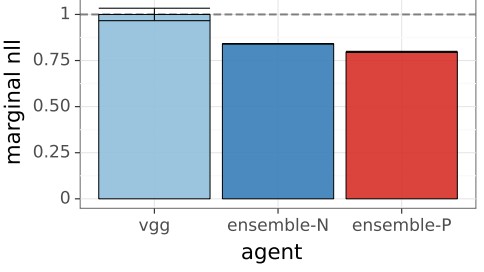 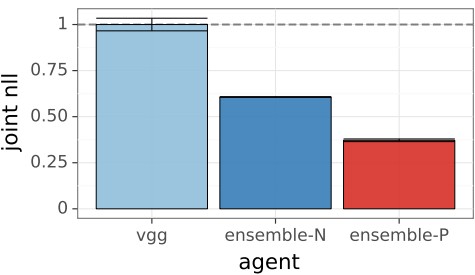

Figure 3: Prior functions significantly improve ensemble agent's joint predictions.

In Figure 3, we observe that prior functions significantly improve an ensemble agent's joint predictions. In Figure 4, We compare the `ensemble-BP` agents along with other ensemble agents and show that bootstrapping doesn't improve the performance over well tuned `ensemble-P` agent on CIFAR10 problems. We use an ensemble of size 10 for all the agents, average the results over 5 random seeds, and normalize the performance w.r.t `ensemble-N` agent. For `ensemble-P` and `ensemble-BP` agents prior scale is swept over $\{1, 3, 10, 30\}$ and the best prior scale is chosen per problem. We observe that all the ensemble agents perform pretty similarly on marginal predictions, except for `ensemble-BP`(0.5). On comparing the agents on joint predictions, we can see that prior functions offer a clear advantage and bootstrapping doesn't offer an advantage over a well tuned `ensemble-P` agent. This is consistent with our observations from Appendix B.4. We suspect that the poor performance of `ensemble-BP`(0.5) on both marginal and joint predictions could be due to the CIFAR10 problems being high-SNR problems and the number of models in each ensemble is small. This is consistent with observations of Fort et al. (2019) that bootstrapping might hurt performance on marginal predictions. Also comparing `ensemble-N` and `ensemble-P` in Figure 3 and 4, we can see that difference between the agents increases as we have more models in the ensemble.

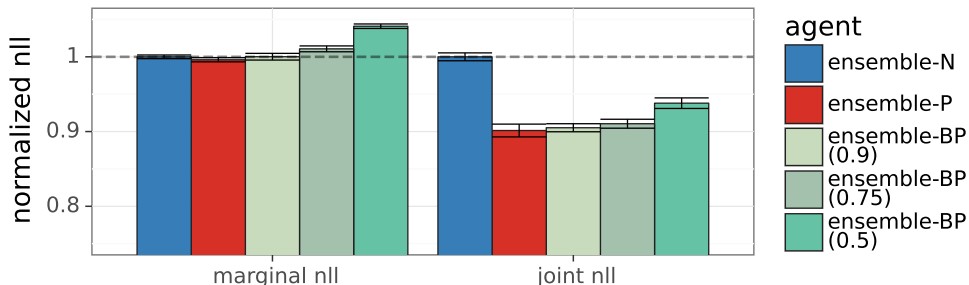

Figure 4: Performance of ensemble agents with 10 models across CIFAR10 problems. Bootstrapping doesn't offer an advantage over well tuned `ensemble-P` in this setting.

In this section, we also compare ensemble agents on problem with non-uniform SNR across input space, similar to Figure 2 in Section 5.2. We follow a similar procedure as Section 5.2 and flip labels of a fraction of data. For this experiment, we consider the full CIFAR10 training dataset and flip the labels of 25% of

randomly picked images corresponding to classes $\{0, 1, 2, 3, 4\}$ and assign each of them to a random uniformly sampled class from $\{5, 6, 7, 8, 9\}$ per image. This creates different SNRs across different classes. Figure 5 shows the results in such setting. All the ensemble agents use 10 models, results are averaged across 5 random seeds, and the performance is normalized w.r.t performance of `ensemble-N`. Best prior scale is chosen for `ensemble-P` and `ensemble-BP` by sweeping over $\{0, 0.3, 1, 3, 10\}$. The performance of `ensemble-P` is very close to that of `ensemble-N`, this might be due to prior functions, based on the randomly initialized convolution networks, being ineffective in this setting with heteroskedasticity. We conjecture that with an appropriate prior which reflects this class imbalance, `ensemble-P` could perform better than `ensemble-N`. We can see that `ensemble-BP`(0.9) and `ensemble-BP`(0.75) offers a clear advantage on joint predictions, and `ensemble-BP`(0.9) offers an improvement even on marginal predictions. The results in Figures 5 and 2 show that when SNR is non-uniform across input space, ensemble agents could further benefit from bootstrapping, even the ensemble agents already use prior functions.

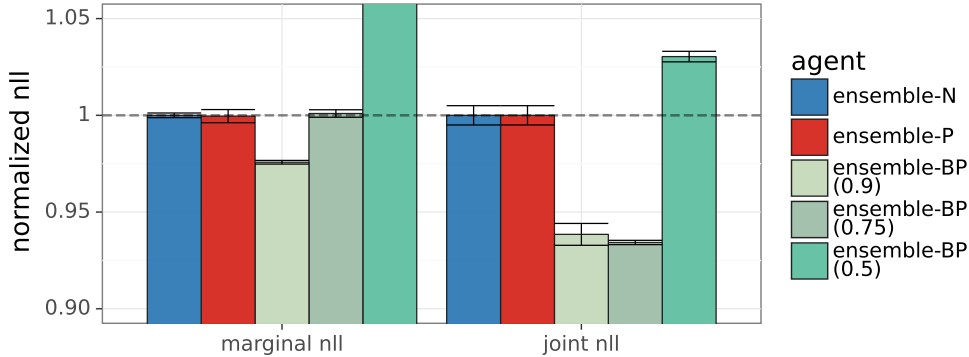

Figure 5: Bootstrapping further improves ensemble agent's joint predictions.

## 6 Bandit experiments

One of our motivations for examining variants of ensemble agents is to understand how to train an ensemble agent that drives effective decision making. The results of Sections 4 and 5 show that prior functions and bootstrapping can improve the quality of joint predictions of ensembles on fixed dataset problems. Theoretically, better joint predictions should enable a learning agent to make better sequential decisions (Wen et al., 2021). We evaluate this empirically using Thompson sampling on heteroscedastic linear bandit problems. Note that this is a relatively simple setting and mainly for illustrative purposes. We leave more extensive experimental work on neural bandits and deep reinforcement learning to future work.

The heteroscedastic linear bandit is a straightforward extension of the heteroscedastic linear regression problem considered in Section 4. First, we sample a set of $N$ actions $\mathcal{X} = \{x_1, \ldots, x_N\}$ i.i.d. from a $d$-dimensional standard normal distribution. We then sample a $d$-dimensional parameter vector $\theta^*$ from a prior $\mathcal{N}(0, \sigma_0^2 I)$, and a diagonal matrix $\Sigma_{obs}$ whose entries are sampled i.i.d from $\mathrm{Uniform}(0, 1)$. The $\Sigma_{obs}$ controls the variance of the observation noise, and for an input $X_t$, the observation noise is given by $W_{t+1} \sim \mathcal{N}(0, X_t^\top \Sigma_{obs} X_t)$. This leads to different actions in $\mathcal{X}$ having difference observation noise variances and hence leading to heteroscedastic behaviour.

At each time $t$, the agent samples a model from the approximate posterior and acts greedily w.r.t the parameter obtained. The agent selects an action $X_t \in \mathcal{X}$ and observes a reward $R_{t+1} = Y_{t+1} = \theta_*^\top X_t + W_{t+1}$. Let $\overline{R}_x = \mathbb{E}[R_{t+1}|\mathcal{E}, X_t = x]$ denote the expected reward of action $x$ conditioned on the environment, and let $X_* \in \arg\max_{x \in \mathcal{X}} \overline{R}_x$ denote the optimal action. We assess agent performance through

$$\mathtt{regret}(T) := \sum_{t=0}^{T-1} \mathbb{E}\left[\overline{R}_{X_*} - \overline{R}_{X_t}\right],$$

which measures the shortfall in expected cumulative rewards relative to an optimal decision maker. Note that the expectation in the regret definition is taken over all the random instances.

In this section, we empirically examine the performance of Thompson sampling (TS) agents, that use variants of ensembles to approximate the posterior over the environment and drive decisions. At each timestep, a TS agent samples an imaginary environment from its approximate posterior and selects an action greedy to the sampled environment (Thompson, 1933; Russo et al., 2018). This simple heuristic is known to effectively balance the exploration and exploitation trade-off, and allows us to evaluate the ability of an ensemble agent to drive decisions. The complete procedure for evaluating agents in bandit problems is provided in Algorithm 1 in Appendix E.

Figure 6 compares the `ensemble-N`, `ensemble-P` and `ensemble-BP` on a simple bandit problem with 2-dimensional features, 4 actions $N = 4$. For all three agents, we consider an infinitely large ensemble, which is equivalent to resolving the optimization problem in (4) with new prior function and data pair weights at each time. This allows us to use the distributions derived in Appendix B.1 for different ensemble agents. We tune parameters of ensemble agents, as outlined in Table 2, to minimize the final regret. Even in this simple problem, we can see that `ensemble-N` performs far worse than other two agents, `ensemble-P` performs much better than `ensemble-N`, but still worse than `ensemble-BP`. This clearly highlights the benefits of prior functions and bootstrapping for ensemble agents.

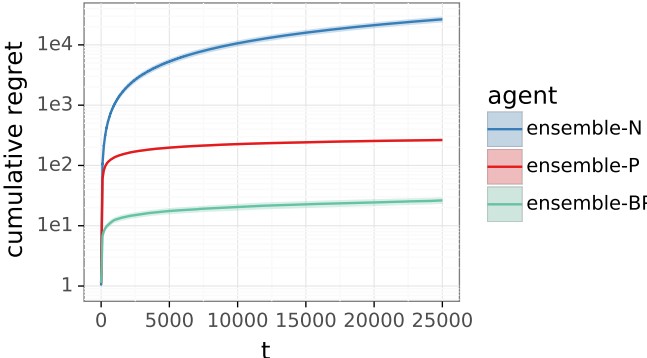

Figure 6: Cumulative regret of ensemble agents on heteroscedastic bandit

## 7 Conclusion

We address a long-standing debate on the efficacy of prior functions and bootstrapping, in ensemble agent training, for uncertainty estimation and decision making. We show that prior functions can significantly enhance the joint predictions of ensemble agents. Moreover, we show that bootstrapping can further improve the performance when SNR varies substantially across the input space. We also show that, if the SNR is uniform across the input space and prior functions are appropriately tuned then bootstrapping offers no significant benefit. We justify our claims using theoretical and experimental results on linear regression, classification, and bandit problems. We believe that some of our analysis for linear regression can also be extended to neural network regression via neural tangent kernel (NTK) type analysis (Jacot et al., 2018; Lee et al., 2019), but we defer this for future. We also believe that many of our claims extend to reinforcement learning problems. Through this work we hope to improve the community's understanding of how to train better ensemble agents for uncertainty estimation.

It is worth emphasizing that most results in this paper are obtained under the assumption that the prior distribution is isometric. As we have discussed, this assumption is without loss of generality since in many problems it can be satisfied after appropriate data pre-processing. In more general settings, we can show that bootstrapping can be helpful in addition to prior functions if SNR varies significantly across the input space, or the covariance matrices of the prior and/or the input distribution are ill-conditioned.

Finally, though this paper has focused on ensemble agents, we believe that the obtained insights on the benefits of prior functions and bootstrapping also extend to other uncertainty estimation approaches, such as

hypermodels (Dwaracherla et al., 2020) and epinets (Osband et al., 2021). We believe this is an interesting direction for future research.

## Acknowledgments

We thank Yee Whye Teh and Brendan O'Donoghue for helpful discussions and feedback. We also thank John Maggs for organization and management of this research effort.

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

# Appendices

## A  Evaluation metrics

Most work on supervised learning has focused on producing accurate *marginal* predictions for each input. However, some recent papers (Wen et al., 2021; Wang et al., 2021; Osband et al., 2022a;b) show that for a broad class of downstream tasks such as combinatorial decision problems and sequential decision problems, accurate marginal predictions are insufficient and accurate *joint* predictions are essential. Wen et al. (2021); Osband et al. (2022a) also propose to evaluate the predictive distributions based on expected KL divergence. As defined in Section 3,

$$\hat{P}_{T:T+\tau-1} = \mathbb{P}(\hat{Y}_{T+1:T+\tau} \in \cdot | \Theta_T, X_{T:T+\tau-1})$$

is the ensemble agent's $\tau$-th order predictive distribution at $\tau$ inputs $X_{T:T+\tau-1}$, and

$$P^*_{T:T+\tau-1} = \mathbb{P}\left(Y_{T+1:T+\tau} \in \cdot | \mathcal{E}, X_{T:T+\tau-1}\right)$$

is the environment's $\tau$-th order predictive distribution at the same inputs. Osband et al. (2022a) proposes to evaluate

$$\mathbf{d}^\tau_{\mathrm{KL}} = \mathbb{E}\left[\mathbf{d}_{\mathrm{KL}}\left(P^*_{T:T+\tau-1}\middle\|\hat{P}_{T:T+\tau-1}\right)\right], \tag{10}$$

where the expectation is over all random variables, including environment $\mathcal{E}$, the ensemble agent parameters $\Theta_T$, inputs $X_{T:T+\tau-1}$ and outcomes $Y_{T+1:T+\tau}$. Recall that $X_{T:T+\tau-1} \equiv (X_T, X_{T+1}, \ldots, X_{T+\tau-1})$ are part of the sequence of data pairs $((X_t, Y_{t+1}) : t = 0, 1, 2, \ldots)$. In many supervised learning problems, they are sampled i.i.d. from an input distribution $P_X$. Also recall that when $\tau = 1$, $\mathbf{d}^\tau_{\mathrm{KL}}$ measures the agent's marginal predictive distribution; when $\tau > 1$, it measures the agent's joint predictive distribution.

### A.1  Metric for Bayesian linear regression

We now discuss the relationship between $\mathbf{d}^\tau_{\mathrm{KL}}$ and the evaluation metric (6) used in Section 4. We explain why the evaluation metric (6) is an equivalent metric of $\mathbf{d}^\tau_{\mathrm{KL}}$ for ensemble agents with a large number of models, as $\tau \to \infty$. The explanation here is informal, and we leave rigorous mathematical proof to future work. First, we also define

$$\overline{P}_{T:T+\tau-1} = \mathbb{P}\left(Y_{T+1:T+\tau} \in \cdot | \mathcal{D}_T, X_{T:T+\tau-1}\right).$$

In a sense, $\overline{P}_{T:T+\tau-1}$ represents the result of perfect inference. We define the expected KL divergence between $\overline{P}_{T:T+\tau-1}$ and $\hat{P}_{T:T+\tau-1}$ as

$$\overline{\mathbf{d}}^\tau_{\mathrm{KL}} = \mathbb{E}\left[\mathbf{d}_{\mathrm{KL}}\left(\overline{P}_{T:T+\tau-1}\middle\|\hat{P}_{T:T+\tau-1}\right)\right].$$

Note that from Proposition 1 of Lu et al. (2021),

$$\mathbf{d}^\tau_{\mathrm{KL}} = \overline{\mathbf{d}}^\tau_{\mathrm{KL}} + \mathbb{I}\left(\mathcal{E}; Y_{T+1:T+\tau} \mid \mathcal{D}_T, X_{T:T+\tau-1}\right),$$

where $\mathbb{I}\left(\mathcal{E}; Y_{T+1:T+\tau} \mid \mathcal{D}_T, X_{T:T+\tau-1}\right)$ is the conditional mutual information between $\mathcal{E}$ and $Y_{T+1:T+\tau}$, conditioned on $\mathcal{D}_T$ and $X_{T:T+\tau-1}$. Note that $\mathbb{I}\left(\mathcal{E}; Y_{T+1:T+\tau} \mid \mathcal{D}_T, X_{T:T+\tau-1}\right)$ does not depend on the agents, thus $\mathbf{d}^\tau_{\mathrm{KL}}$ and $\overline{\mathbf{d}}^\tau_{\mathrm{KL}}$ are equivalent for agent comparison.

Recall that for the Bayesian linear regression problem described in Section 4, $\Theta_T$ includes $M$ samples drawn i.i.d. from $\mathbb{P}\left(\hat{\theta}_m \in \cdot \middle| \mathcal{D}_T\right) = N\left(\hat{\mu}_T, \hat{\Sigma}_T\right)$. When $M$ is sufficiently large, we have

$$\mathbb{P}\left(\hat{Y}_{T+1:T+\tau} \in \cdot \middle| \Theta_T, X_{T:T+\tau-1}\right) \approx \mathbb{P}\left(\hat{Y}_{T+1:T+\tau} \in \cdot \middle| \hat{\mu}_T, \hat{\Sigma}_T, X_{T:T+\tau-1}\right).$$

On the other hand, recall the posterior over $\theta_*$, which indexes the environment $\mathcal{E}$, is $N\left(\mu_T, \Sigma_T\right)$. Hence, $\overline{P}_{T:T+\tau-1}$ can be rewritten as

$$\mathbb{P}\left(Y_{T+1:T+\tau} \in \cdot | \mu_T, \Sigma_T, X_{T:T+\tau-1}\right).$$

Consequently, when $M$ is sufficiently large, we have

$$\mathbf{d}_{\mathrm{KL}}^{\tau} \approx \mathbb{E}\left[\mathbf{d}_{\mathrm{KL}}\left(\mathbb{P}\left(Y_{T+1:T+\tau} \in \cdot | \mu_T, \Sigma_T, X_{T:T+\tau-1}\right) \middle\| \mathbb{P}\left(\hat{Y}_{T+1:T+\tau} \in \cdot \middle| \hat{\mu}_T, \hat{\Sigma}_T, X_{T:T+\tau-1}\right)\right)\right].$$

Finally, as is discussed in Section 2.5 of Wen et al. (2021), as $\tau \to \infty$, under suitable regularity conditions, we have

$$\begin{aligned}
\mathbf{d}_{\mathrm{KL}}^{\tau} &\approx \mathbb{E}\left[\mathbf{d}_{\mathrm{KL}}\left(N(\mu_T, \Sigma_T) \middle\| N(\hat{\mu}_T, \hat{\Sigma}_T)\right)\right] \\
&= \mathbb{E}\left[\mathbf{d}_{\mathrm{KL}}\left(\mathbb{P}\left(\theta_* \in \cdot \middle| \mathcal{D}_T\right) \middle\| \mathbb{P}\left(\hat{\theta}_m \in \cdot \middle| \mathcal{D}_T\right)\right)\right].
\end{aligned} \tag{11}$$

In other words, as $\tau \to \infty$, $\mathbf{d}_{\mathrm{KL}}^{\tau}$ is approximately equivalent to the metric in (6) for comparing ensemble agents with a large number of models.

## A.2 Dyadic sampling

One limitation of $\mathbf{d}_{\mathrm{KL}}^{\tau}$ is that the magnitude of $\tau$ required to provide additional insight beyond marginals can become intractably large, especially when the dimension of the input $X_t$ is high (Osband et al., 2022b). As a solution, Osband et al. (2022b) proposes to use dyadic sampling to evaluate joint predictive distributions. In a high level, dyadic sampling changes the joint distribution of inputs $X_{T:T+\tau-1}$: it first samples two random *anchor points* from the input space, and then resamples $\tau$ inputs from these two anchor points. The expected KL-divergence under dyadic sampling is referred to as $\mathbf{d}_{\mathrm{KL}}^{\tau,2}$.

Similar to Osband et al. (2022b), we use dyadic sampling with $\tau = 10$ to evaluate ensemble agents' joint predictions in Section 5. Note that $\mathbf{d}_{\mathrm{KL}}^{\tau,2}$ with $\tau = 10$ can be efficiently computed based on Monte-Carlo simulation (see Algorithm 1 of Osband et al. (2022b)).

## A.3 Expected KL-divergence and negative log-likelihoods

As discussed in recent papers (Wen et al., 2021; Osband et al., 2022b), $\mathbf{d}_{\mathrm{KL}}^{\tau}$ and its variants (e.g. $\mathbf{d}_{\mathrm{KL}}^{\tau,2}$) are equivalent to variants of negative log-likelihood (nll), which is also known as the *cross-entropy loss*, for agent comparison. In particular, note that the $\tau$-th order cross-entropy loss is

$$\mathbf{d}_{\mathrm{CE}}^{\tau} \equiv -\mathbb{E}\left[\log \hat{P}_{T:T+\tau-1}(Y_{T+1:T+\tau})\right], \tag{12}$$

and it is straightforward to show that

$$\mathbf{d}_{\mathrm{KL}}^{\tau} = \mathbf{d}_{\mathrm{CE}}^{\tau} + \mathbb{E}[\log P^*_{T:T+\tau-1}(Y_{T+1:T+\tau})].$$

Since $P^*_{T:T+\tau-1}(Y_{T+1:T+\tau})$ does not depend on the agent, $\mathbf{d}_{\mathrm{KL}}^{\tau}$ and $\mathbf{d}_{\mathrm{CE}}^{\tau}$ rank agents identically.

For experiments on the CIFAR10 dataset, since $P^*_{T:T+\tau-1}(Y_{T+1:T+\tau})$, the likelihood under the true environment $\mathcal{E}$, is unknown, we choose to use $\tau$-th order cross-entropy loss or negative log-likelihood (nll) to compare agents.

# B Bayesian linear regression

## B.1 Perturbed loss function and solution

As is described in Section 4, the perturbed loss function is:

$$\hat{\theta}_m \in \arg\min_{\theta} \sum_{t=0}^{T-1} \frac{\nu(X_t)}{2}\left(\theta^T X_t - [Y_{t+1} + Z_{t+1,m}]\right)^2 + \frac{\lambda}{2}\left\|\theta - \tilde{\theta}_m\right\|_2^2,$$

where $Z_{t+1,m}$ is additive perturbation for data pair $(X_t, Y_{t+1})$ sampled from $N\left(0, \hat{\sigma}^2(X_t)\right)$; $\nu, \hat{\sigma}^2 : \Re^d \to \Re^+$ are respectively the weight function and the bootstrapping variance function; $\lambda \geq 0$ is the regularization

coefficient; and $\tilde{\theta}_m$ is independently sampled from $N(0, \hat{\sigma}_0^2 I)$. The magnitude of $\hat{\sigma}_0^2$ reflects the agent's prior uncertainty, and hence $\hat{\sigma}_0$ is known as the agent's *prior scale*. Note that $\|\theta - \tilde{\theta}_m\|_2^2$ is a randomly perturbed regularizer, and is equivalent to a prior function for linear regression (Osband et al., 2018). It is straightforward to show that

$$\hat{\theta}_m = \left( \sum_{t=0}^{T-1} \nu(X_t) X_t X_t^T + \lambda I \right)^{-1} \left( \sum_{t=0}^{T-1} \nu(X_t) \left( Y_{t+1} + Z_{t+1,m} \right) X_t + \lambda \tilde{\theta}_m \right).$$

Note that $\hat{\theta}_m \mid \mathcal{D}_T \sim N\left( \hat{\mu}_T, \hat{\Sigma}_T \right)$, where

$$\hat{\mu}_T = \left( \sum_{t=0}^{T-1} \nu(X_t) X_t X_t^T + \lambda I \right)^{-1} \left( \sum_{t=0}^{T-1} \nu(X_t) Y_{t+1} X_t \right)$$

$$\hat{\Sigma}_T = \left( \sum_{t=0}^{T-1} \nu(X_t) X_t X_t^T + \lambda I \right)^{-1} \left( \sum_{t=0}^{T-1} \nu^2(X_t) \hat{\sigma}^2(X_t) X_t X_t^T + \lambda^2 \hat{\sigma}_0^2 I \right) \left( \sum_{t=0}^{T-1} \nu(X_t) X_t X_t^T + \lambda I \right)^{-1}.$$

### B.2 Proofs for Proposition 1 and 2

#### Proof for Proposition 1

**Proof** Based on the analytical formula of $\hat{\mu}_T$ and $\hat{\Sigma}_T$ provided in Appendix B.1, if $\hat{\sigma}^2(\cdot) = \sigma^2(\cdot)$, $\nu(\cdot) = 1/\sigma^2(\cdot)$, $\hat{\sigma}_0^2 = \sigma_0^2$, and $\lambda = 1/\sigma_0^2$, we have $\hat{\mu}_T = \mu_T$ and $\hat{\Sigma}_T = \Sigma_T$. Thus, $\mathbb{P}(\hat{\theta}_m \in \cdot \mid \mathcal{D}_T) = \mathbb{P}(\theta_* \in \cdot \mid \mathcal{D}_T)$. ∎

#### Proof for Proposition 2

**Proof** Note that when $\hat{\sigma}^2(\cdot) = 0$ and $\hat{\sigma}_0^2 = 0$, we have

$$\hat{\theta}_m = \left( \sum_{t=0}^{T-1} \nu(X_t) X_t X_t^T + \lambda I \right)^{-1} \left( \sum_{t=0}^{T-1} \nu(X_t) Y_{t+1} X_t \right),$$

which is deterministic conditioned on $\mathcal{D}_T$. ∎

### B.3 Proof for Theorem 1

**Proof** Notice that to ensure that $\hat{\mu}_T = \mu_T$, we need $\nu(\cdot) = c/\sigma^2(\cdot)$ and $\lambda = c/\sigma_0^2$ for a constant $c > 0$. Without loss of generality, we choose $c = 1$. Under this choice and $\hat{\sigma}^2(\cdot) = 0$, we have:

$$\hat{\Sigma}_T = \left( \sum_{t=0}^{T-1} \frac{1}{\sigma^2(X_t)} X_t X_t^T + \frac{1}{\sigma_0^2} I \right)^{-1} (\hat{\sigma}_0^2/\sigma_0^4 I) \left( \sum_{t=0}^{T-1} \frac{1}{\sigma^2(X_t)} X_t X_t^T + \frac{1}{\sigma_0^2} I \right)^{-1}$$

$$= \underbrace{\hat{\sigma}_0^2/\sigma_0^4}_{=\eta} \Sigma_T^2 = \eta \Sigma_T^2, \tag{13}$$

where we define the shorthand notation $\eta = \hat{\sigma}_0^2/\sigma_0^4$ to simplify the exposition. Then following the KL-divergence formula for multivariate Gaussian distributions, we have

$$\mathbf{d}_{\mathrm{KL}}\left( P\left( \theta_* \in \cdot \mid \mathcal{D}_T \right) \,\Big\|\, P\left( \hat{\theta}_m \in \cdot \mid \mathcal{D}_T \right) \right) = \mathbf{d}_{\mathrm{KL}}\left( N(\mu_T, \Sigma_T) \,\|\, N(\mu_T, \eta \Sigma_T^2) \right)$$

$$= \frac{1}{2} \left[ d \log \eta + \log \det(\Sigma_T) - d + \frac{1}{\eta} \mathrm{tr}\left[ \Sigma_T^{-1} \right] \right]. \tag{14}$$

Taking an expectation over $\mathcal{D}_T$, we have

$$
\begin{aligned}
\mathbb{E}\left[\mathbf{d}_{\mathrm{KL}}\left(P\left(\theta_* \in \cdot | \mathcal{D}_T\right) \middle\| P\left(\hat{\theta}_m \in \cdot | \mathcal{D}_T\right)\right)\right] &= \frac{1}{2}\mathbb{E}\left[d\log\eta + \log\det\left(\Sigma_T\right) - d + \frac{1}{\eta}\operatorname{tr}\left[\Sigma_T^{-1}\right]\right] \\
&\overset{(a)}{=} \frac{1}{2}\mathbb{E}\left[d\log\eta - \log\det\left(\Sigma_T^{-1}\right) - d + \frac{1}{\eta}\operatorname{tr}\left[\Sigma_T^{-1}\right]\right] \\
&= \frac{1}{2}\left[d\log\eta - \mathbb{E}\left[\log\det\left(\Sigma_T^{-1}\right)\right] - d + \frac{1}{\eta}\mathbb{E}\left[\operatorname{tr}\left[\Sigma_T^{-1}\right]\right]\right] \\
&\overset{(b)}{\geq} \frac{1}{2}\left[d\log\eta - \log\det\left(\mathbb{E}\left[\Sigma_T^{-1}\right]\right) - d + \frac{1}{\eta}\operatorname{tr}\left[\mathbb{E}\left[\Sigma_T^{-1}\right]\right]\right], \quad (15)
\end{aligned}
$$

where (a) follows from $\log\det\left(\Sigma_T\right) = -\log\det\left(\Sigma_T^{-1}\right)$, and (b) follows from $\mathbb{E}\left[\operatorname{tr}\left[\Sigma_T^{-1}\right]\right] = \operatorname{tr}\left[\mathbb{E}\left[\Sigma_T^{-1}\right]\right]$ and

$$
\mathbb{E}\left[\log\det\left(\Sigma_T^{-1}\right)\right] \leq \log\det\left(\mathbb{E}\left[\Sigma_T^{-1}\right]\right)
$$

since $\log\det(\cdot)$ is a concave function. Note that

$$
\Sigma_T^{-1} = \frac{1}{\sigma_0^2}\left[I + \sum_{t=0}^{T-1} \frac{\sigma_0^2 X_t X_t^T}{\sigma^2(X_t)}\right],
$$

hence

$$
\mathbb{E}\left[\Sigma_T^{-1}\right] = \frac{1}{\sigma_0^2}\left[I + T\Gamma\right].
$$

Let $\zeta_1, \zeta_2, \ldots \zeta_d$ denote the eigenvalues of $\mathbb{E}\left[\Sigma_T^{-1}\right]$, consequently we have

$$
\zeta_i = \frac{1 + T\gamma_i}{\sigma_0^2}.
$$

where $\gamma_i$'s are eigenvalues of $\Gamma = \mathbb{E}\left[\frac{\sigma_0^2 X_t X_t^T}{\sigma^2(X_t)}\right]$. By rewriting $\log\det\left(\mathbb{E}\left[\Sigma_T^{-1}\right]\right)$ and $\operatorname{tr}\left[\mathbb{E}\left[\Sigma_T^{-1}\right]\right]$ using eigenvalues, we have

$$
\mathbb{E}\left[\mathbf{d}_{\mathrm{KL}}\left(P\left(\theta_* \in \cdot | \mathcal{D}_T\right) \middle\| P\left(\hat{\theta}_m \in \cdot | \mathcal{D}_T\right)\right)\right] \geq \frac{1}{2}\left[d\log\eta - \sum_{i=1}^{d}\log\zeta_i - d + \frac{1}{\eta}\sum_{i=1}^{d}\zeta_i\right].
$$

By minimizing the right-hand side of the above inequality over $\eta$, which is equivalent to minimizing over the prior sample variance $\hat{\sigma}_0^2$, we obtain the minimizing $\eta$

$$
\eta_* = \frac{1}{d}\sum_{i=1}^{d}\zeta_i.
$$

Plug in the minimizing $\eta_*$, we have

$$
\mathbb{E}\left[\mathbf{d}_{\mathrm{KL}}\left(P\left(\theta_* \in \cdot | \mathcal{D}_T\right) \middle\| P\left(\hat{\theta}_m \in \cdot | \mathcal{D}_T\right)\right)\right] \geq \frac{1}{2}\left[d\log\left(\frac{1}{d}\sum_{i=1}^{d}\zeta_i\right) - \sum_{i=1}^{d}\log\zeta_i\right].
$$

Rewrite the above equation using the eigenvalues of $\Sigma_X$, we have

$$
\begin{aligned}
\mathbb{E}\left[\mathbf{d}_{\mathrm{KL}}\left(P\left(\theta_* \in \cdot | \mathcal{D}_T\right) \middle\| P\left(\hat{\theta}_m \in \cdot | \mathcal{D}_T\right)\right)\right] &\geq \frac{1}{2}\left[d\log\left(\frac{1 + T\bar{\gamma}}{\sigma_0^2}\right) - \sum_{i=1}^{d}\log\left(\frac{1 + T\gamma_i}{\sigma_0^2}\right)\right] \\
&= \frac{1}{2}\sum_{i=1}^{d}\ln\left(\frac{1 + T\bar{\gamma}}{1 + T\gamma_i}\right), \quad (16)
\end{aligned}
$$

where $\bar{\gamma} = \frac{1}{d}\sum_{i=1}^{d}\gamma_i$. This concludes the proof. ∎

## B.4 When bootstrapping is not strictly necessary

We would like to clarify that bootstrapping is not always strictly necessary for the considered Bayesian linear regression problems, and there are cases in which `ensemble-P` can work well. The following is one example: assume that $X_t$'s are independently drawn from $N(0, I)$, and the linear regression problem is homoscedastic in the sense that $\sigma^2(X_t) = \sigma^2$ is a constant. In this case, if `ensemble-P` chooses $\nu(X_t) = 1/\sigma^2$ for all $X_t$ and $\lambda = 1/\sigma_0^2$, then we have $\hat{\mu}_T = \mu_T$,

$$\Sigma_T = \left[I/\sigma_0^2 + \sum_{t=0}^{T-1} X_t X_t^T/\sigma^2\right]^{-1} \quad \text{and} \quad \hat{\Sigma}_T = \hat{\sigma}_0^2 \Sigma_T^2/\sigma_0^4$$

(see Appendix B.1). Also note that for sufficiently large $T$, $\frac{1}{T}\sum_{t=0}^{T-1} X_t X_t^T \approx I$ due to Law of large numbers. Hence, we have $\Sigma_T \approx \frac{\sigma_0^2\sigma^2}{\sigma^2+T\sigma_0^2}I$ and $\hat{\Sigma}_T \approx \frac{\hat{\sigma}_0^2\sigma^4}{(\sigma^2+T\sigma_0^2)^2}I$. In other words, if we choose

$$\hat{\sigma}_0^2 = \left(1 + T\sigma_0^2/\sigma^2\right)\sigma_0^2,$$

then $P\left(\hat{\theta}_m \in \cdot \big| \mathcal{D}_T\right)$ approximately matches $P\left(\theta_* \in \cdot \big| \mathcal{D}_T\right)$ under `ensemble-P`. Notice that this choice of $\hat{\sigma}_0^2$ depends on the training data size $T$. Also note that for this case, the lower bound in Theorem 1 is zero since $\Gamma = \sigma_0^2/\sigma^2 I$.

Another example is a high SNR scenario where $\sigma^2(X_t) \approx 0$ for all input $X_t$. From Proposition 1, the optimal `ensemble-BP` agent will choose $\hat{\sigma}^2(X_t) = \sigma^2(X_t) \approx 0$ for all $X_t$. Consequently, under mild technical conditions, an `ensemble-P` agent can be near-optimal.

## C  Neural Testbed experiments

Neural Testbed (Osband et al., 2022a) is an open-source test suite designed to evaluate the predictive distributions of deep learning algorithms on randomly generated classification problems, based on a neural network generative model. Problem instances are binary classification problems which are sampled from a random 2-layer MLP, with 50 hidden units, across different input dimensions $d \in \{2, 10, 100\}$, training set sizes $T = dr$ with $r \in \{1, 10, 100, 1000\}$, and output temperatures $\{0.01, 0.1, 0.5\}$ which correspond to high, medium, and low SNR. By default, results are averaged over 10 random seeds in each setting, and confidence bars are also reported. We evaluate the agents based on their marginal predictions and joint predictions as discussed in Appendix A.

### C.1  Agent description

The `mlp` agent uses a single point estimate that has the same network architecture as the generative model, which is a 2-hidden layer MLP with 50 units in each hidden layer. The `ensemble-N` agent uses 100 models with each model being a 2-layer MLP that only differ in initialization. The `ensemble-P` agent also uses 100 models with each model being a 2-layer MLP with an additive prior function which is also a random 2-layer MLP (Osband et al., 2018). In specific, $m$-th model of the `ensemble-P` model takes the form

$$f_{\theta_m}(x) = g_{\theta_m}(x) + p_m(x),$$

where $g_{\theta_m}, p_m(x) : \mathcal{X} \to \Re^{|\mathcal{Y}|}$ are the trainable network and randomly initialized prior function which remains fixed through out the experiment. $p_m(x)$ differs across $m$ only through initialization. The `ensemble-BP` also uses 100 models that are similar to models of `ensemble-P`. The main difference between the `ensemble-BP` and `ensemble-P` is the loss function. In Equation (9), `ensemble-N` and `ensemble-P` chooses constant weights $W_{t+1,m} = 1$, $\forall m, t$, while `ensemble-BP` used in Neural Testbed experiments samples $W_{t+1,m}$ i.i.d from Bernoulli(0.5).

We modify the code from enn library at https://github.com/deepmind/enn and Neural Testbed library at https://github.com/deepmind/neural_testbed to model our agents and run experiments on Neural Testbed. We run our experiments using 8-core CPU and 4 GB ram instances on Google cloud compute. In

all the figures, we normalized the scores of all agents in each setting with the score of an uniform random agent, which picks each class 0 and 1, with probability of 0.5. This helps us to ensure that problems across different temperature and input dimensions contribute roughly equally to the final score.

For the ensemble agents used for Neural Testbed experiments, we find the best hyperparamters by sweeping over a set of hyperparameters. We use similar sweeps as mentioned in Osband et al. (2022b). For `mlp` and `ensemble-N` agents, for a problem with input dimension $D$ and temperature $\rho$, we choose the weight decay term ($\lambda$ in Equation 9) from $\lambda \in \{0.1, 0.3, 1, 3, 10\} \times d/\sqrt{\rho}$. For `ensemble-P` agent, in addition to sweeping over the weight decay term, we also sweep over the prior scale of the additive prior functions. In specific, for a problem with temperature $\rho$, we choose values from $\{0.3/\sqrt{\rho}, 0.3/\rho, 1/\sqrt{\rho}, 1/\rho, 3/\sqrt{\rho}, 3/\rho, \}$.

### C.2 Additional results

#### `ensemble-P`'s sensitivity to prior scale:

Figure 7 shows the sensitivity of performance of `ensemble-P` agent with respect to different prior scales. In particular we choose input dimension 100 and temperature of 0.01. But we observe similar trend across other settings as well. We observe that the performance of `ensemble-P` agent does depend on the prior scale, but the performance is not too sensitive as long as prior scale is roughly correct. The dotted line denote the performance of an uniform random agent which picks both classes with equal probability. We can see that, especially when the training dataset size is small, the `ensemble-N` agent (prior scale is 0) performs close to an uniform random agent.

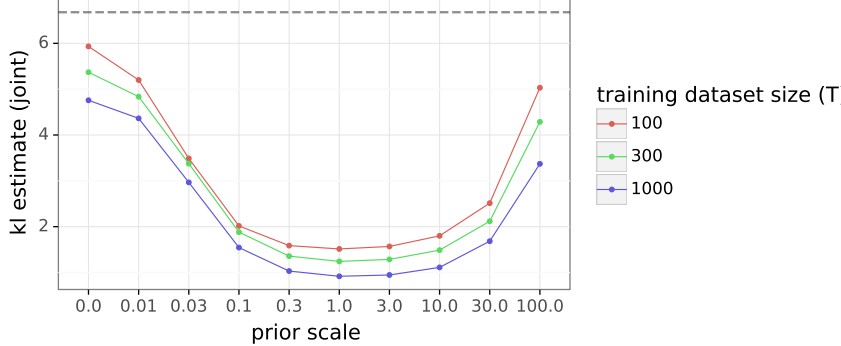

Figure 7: We see that performance does depend on the prior scale, but the performance is not very sensitive to prior scale as long as it is in the correct range. The dotted line indicates the performance of a uniform random agent.

#### Performance of `ensemble-N` on varying SNR testbed problems

Figure 8 is an extension of Figure 2 with the performance of `ensemble-N` included on the varying SNR problems derived by flipping labels of Neural Testbed problems. We can see that all three ensemble agents perform very similarly on marginal predictions. However, when evaluated by joint predictions, `ensemble-P` performs significantly better than `ensemble-N` and, with bootstrapping, `ensemble-BP` improves over `ensemble-P`.

Figure 9 shows the breakdown of performance of the agents in Figure 8 for input dimension 10. The main improvement of `ensemble-BP` over `ensemble-P` arises when the training dataset size $T$ is not too small. When $T$ is close to 0, prior plays the most prominent role, and after a point, as $T$ increases the posterior distribution concentrates and the benefits of bootstrapping start to diminish.

## D Cifar10 experiments

CIFAR10 has been widely known in the deep learning community as a benchmark dataset and used to evaluate classification algorithms. Each data pair in CIFAR10 dataset consists of an image and a label corresponding

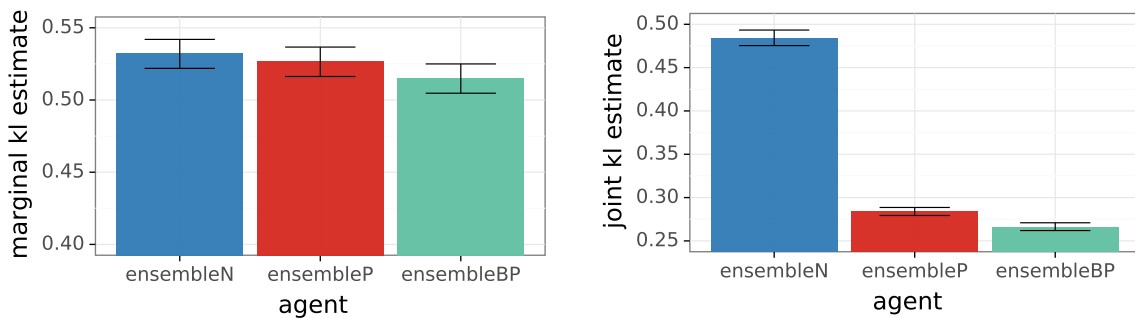

Figure 8: The ensemble agents are distinguished on joint predictions. `ensemble-P` improves over the performance of `ensemble-N` agent which is further improved by `ensemble-BP`.

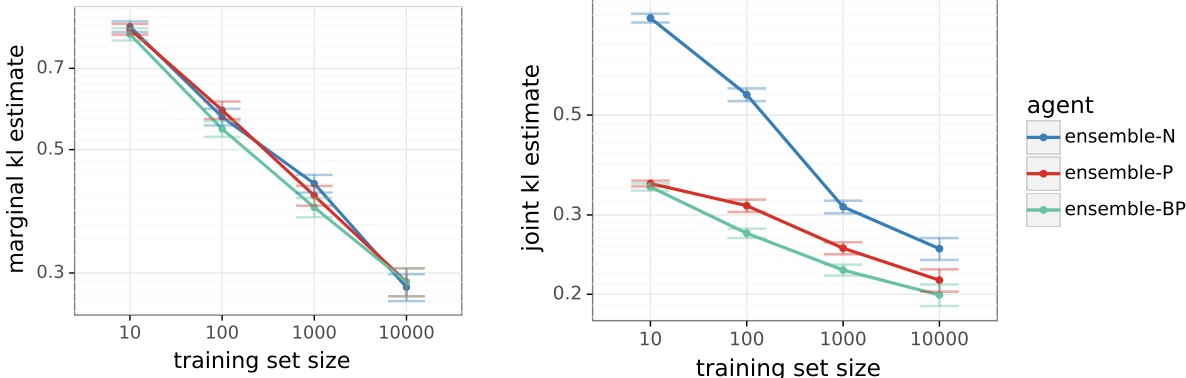

Figure 9: Breakdown of bootstrap results on problems with varying SNR

to one of 10 classes which we refer to as $\{0, 1, 2, \ldots 9\}$. We look at different problems constructed from the dataset, by limiting the training set size. In particular, we consider training sets of sizes $10, 100, 1000$, and $50000$, where $50000$ is the size of the full training dataset.

For Figure 5, we flip the labels of 25% of images corresponding to first 5 classes, $\{0, 1, 2, 3, 4\}$, and assign each of them to a random uniformly sampled class from the other 5 classes, $\{5, 6, 7, 8, 9\}$ per image. This creates different SNRs across different classes. For this experiment, we consider a single setting, where the agents are trained with $50,000$ training samples (full training data).

## D.1    Agent description

For the experiments on CIFAR10 dataset, we use a **vgg** model (Simonyan & Zisserman, 2014) as the single point estimate model. An **ensemble-N** agent consists of an ensemble of **vgg** models and **ensemble-P** agent consists of an ensemble of models with each model being a **vgg** model with output of a small randomly initialized convolution network added to the logits. Each model of **ensemble-BP** is same as that of **ensemble-P**, but uses a slightly different loss function. In Equation (9), **ensemble-N** and **ensemble-P** uses constant weights $W_{t+1,m} = 1, \forall m, t$, while **ensemble-BP** uses $W_{t+1,m}$ sampled i.i.d from Bernoulli distribution. We consider few variants of **ensemble-BP** based on the mean probability of the Bernoulli distribution. We use **ensemble-BP**$(p)$ to denote the **ensemble-BP** agent with weights $W_{t+1,m}$ sampled i.i.d from Bernoulli$(p)/p$.

The single point estimate agent **vgg** uses VGG model (Simonyan & Zisserman, 2014). We make use of the VGG model implementation at `https://github.com/deepmind/enn`. The VGG model consists of 11 blocks whose output is finally passed through an average pool layer and a dense layer to get the logits. Each of the 11 bolcks consists of a convolution layer, batchnorm layer, and a relu layer. The 11 blocks have convolution layers with kernel width of $(3, 3)$ with number of channels from 1st to 11th

block as $(64, 64, 128, 128, 128, 256, 256, 256, 512, 512, 512)$ and stride length as $(1, 1, 2, 1, 1, 2, 1, 1, 2, 1, 1)$. The `ensemble-N` agent uses an ensemble of multiple VGG models. The `ensemble-P` agent uses an ensemble of multiple models, with each model consisting of a VGG model combined with a randomly initialized convolution network at the logits. The convolution prior network consists of 3 convolution layers with kernel of size $(5, 5)$ and output channels as $4, 8, 4$ in first, second, and third layers respectively. Finally the output of third layer is passed through a dense layer to get logits which are added to the logits of VGG model. The `ensemble-BP` agent uses the same network as `ensemble-P`.

For experiments on CIFAR10, we consider 4 different problems, based on the training dataset size. Once trained, all the agents in all problems are evaluated on the same (full) test dataset. We choose the weight decay of $30/($training dataset size$)$ for all the agents. This value was obtained by sweeping weight decay for `ensemble-N` with 10 VGG models. For `ensemble-P` agent in Figure 3 and 10, we choose a prior scale of 30. In the experiments for Figure 11, both `ensemble-P` and `ensemble-BP` use a prior scale of 3. For all ensemble agents, we train the the models of ensemble separately and combine them during evaluation. Each model is trained on 2x2 TPU with per-device batch size of 32. Each model is trained for 400 epochs. For training, we use an SGD optimizer with learning rate schedule with initial learning rate as $\{0.0001, 0.001, 0.01, 0.025\}$ for training dataset sizes $\{10, 100, 1000, 50000\}$ and the learning rate is reduced to one-tenth after 200 epochs, one-hundredth after 300 epochs, and one-thousandth after 350 epochs.

## D.2 Additional results

Figure 3 compares the average performance of `vgg`, `ensemble-N`, and `ensemble-P` across CIFAR10 problems of different training dataset sizes. Error bars are obtained by evaluating the performance of agents across 5 seeds. For each seed, we compute the performance of `ensemble-N` and `ensemble-P` agents, by sampling 100 particles with repetition from a set of 100 trained particles of the corresponding agent. This was done to avoid training a huge number of particles. Figure 10 shows a more detailed breakdown of performance across CIFAR10 problems of different training dataset sizes. We see that on marginal predictions, both `ensemble-N` and `ensemble-P` perform similarly, while they are clearly distinguished on joint predictions. We also see that the benefit of the prior functions is higher when the training dataset size is lower. This is what we expect as the prior information matters the most when we have low data.

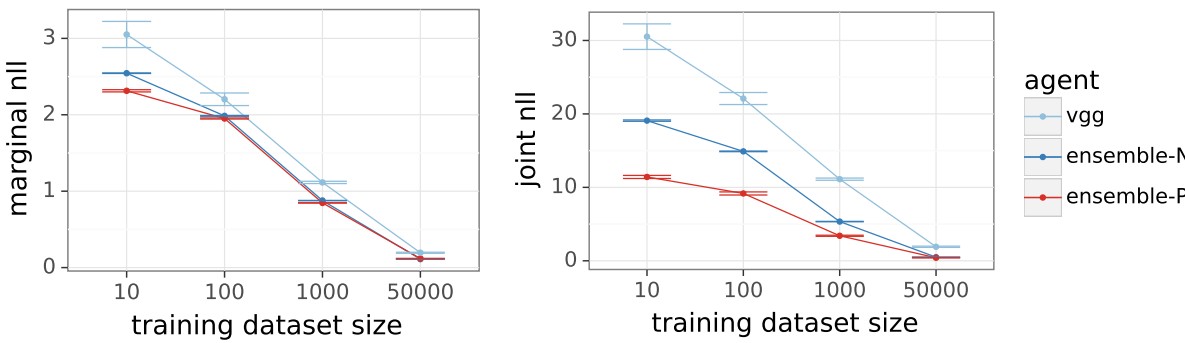

Figure 10: Performance of ensemble agents with 100 models across CIFAR10 problems of different training dataset sizes.

Figure 11 shows the average performance of ensemble agents on CIFAR10 problems across different ensemble sizes. We consider a slight variation of the above problem, where 1% of the images have their labels assigned to uniformly random values in $\{1, 2, \ldots 10\}$. This ensures that the problem is not extremely high SNR. We use a fixed prior scale of 3 for `ensemble-P` and `ensemble-BP` on all the problems. Note that this might not be the best prior scale for `ensemble-P` and `ensemble-BP` agents; however, this helps in gaining insights into bootstrapping methods. The results are normalized w.r.t performance of `ensemble-N`. We see that for small ensemble sizes, on marginal predictions, bootstrapping might actually hurt the performance as suggested in Fort et al. (2019). However, as we increase the ensemble size, even on marginal predictions, bootstrapping

starts to help. On joint predictions, which are actually indicative of downstream tasks (Wen et al., 2021), bootstrapping offers a clear benefit.

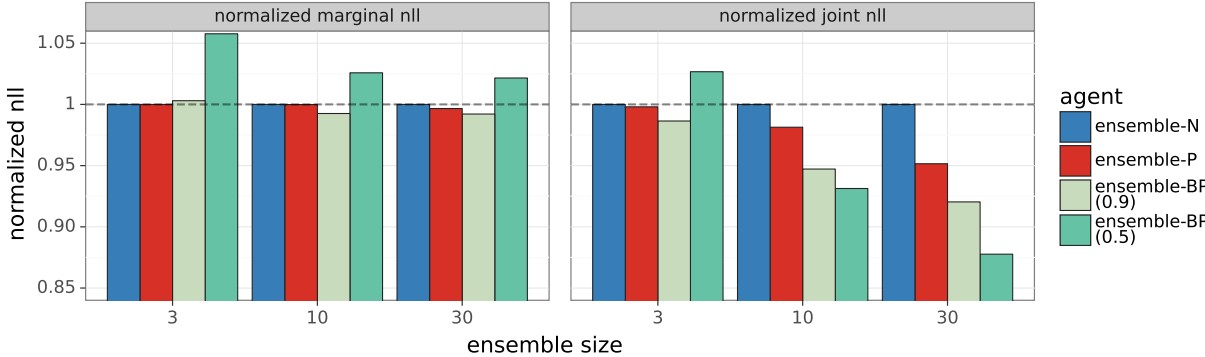

Figure 11: Performance of ensemble agents across different ensemble sizes. Bootstrapping helps on larger ensembles when evaluated on joint predictions.

## E    Bandit experiments

Section 6 demonstrates the performances of variants of ensemble agents, including `ensemble-N`, `ensemble-P`, and `ensemble-BP`, on randomly generated bandit problems. In particular, all the ensemble agents use Thompson sampling (TS) as the exploration scheme. Algorithm 1 provides the pseudo-code for how we evaluate ensemble agents on bandit problems. In particular, it describes how random bandit problems are sampled and how the ensemble agents adaptively choose actions.

Figure 12 is an extension of Figure 6 with an additional agent that is a variation of `ensemble-P` which uses different weights for data pairs based on the observation noise variance at that data pair. Typically, when `ensemble-N` or `ensemble-P` agents are used, they choose a fixed weight across all data pairs. We can see that using weights based on observation noise variance improves performance to some extent, but `ensemble-BP` still performs the best. This further justifies the benefits of bootstrapping.

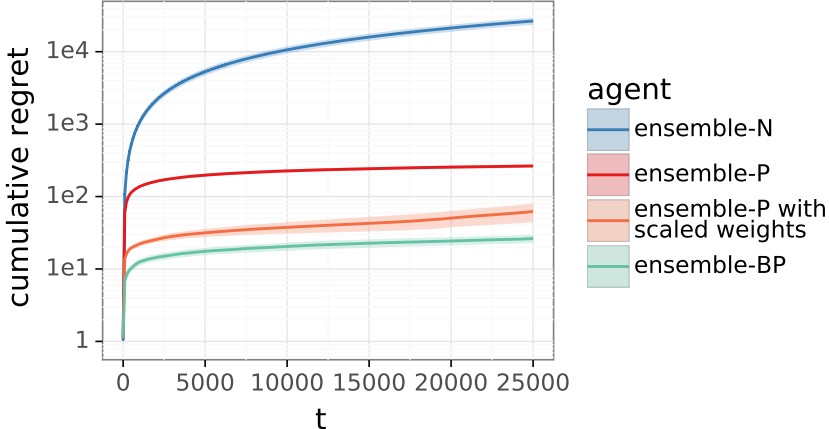

Figure 12: Heteroscadastic linear bandit with 4 actions

---

**Algorithm 1** Evaluation of ensemble agents on bandit problems

---

**Require:** the following inputs are required

1. distribution over the environment $\mathbb{P}(\mathcal{E} \in \cdot)$, action distribution $P_X$, and the number of actions $N$.

2. ensemble agent type (e.g. `ensemble-N`, `ensemble-P`, or `ensemble-BP`)

3. number of time steps $T$

4. number of sampled problems $J$

**for** $j = 1, 2, \ldots, J$ **do**

  **Step 1: sample a bandit problem**
    1. sample environment $\mathcal{E} \sim \mathbb{P}(\mathcal{E} \in \cdot)$
    2. sample a set $\mathcal{X}$ of $N$ actions $x_1, x_2, \ldots, x_N$ i.i.d. from $P_X$
    3. obtain the mean rewards corresponding to actions in $\mathcal{X}$

$$\overline{R}_x = \mathbb{E}[Y_{t+1} \mid \mathcal{E}, X_t = x], \quad \forall x \in \mathcal{X}$$

    4. compute the optimal expected reward $\overline{R}_* = \max_{x \in \mathcal{X}} \overline{R}_x$

  **Step 2: run the ensemble agent**
  Initialize the data buffer $\mathcal{D}_0 = ()$
  **for** $t = 1, 2, \ldots, T$ **do**
    1. Update agent parameters $\Theta_t$ based on $\mathcal{D}_{t-1}$
    2. action selection based on TS:
      i. sample model $\hat{\mathcal{E}}_t$ from the agent belief distribution

$$\hat{\mathcal{E}}_t \sim \mathbb{P}\left(\hat{\mathcal{E}} \in \cdot \mid \Theta_t\right)$$

      ii. act greedily based on $\hat{\mathcal{E}}_t$

$$X_t \in \arg\max_{x \in \mathcal{X}} \mathbb{E}[\hat{Y}_{t+1} = 1 \mid \hat{\mathcal{E}}_t, X_t = x]$$

      iii. generate observation $Y_{t+1}$ based on action $X_t$

$$Y_{t+1} \sim \mathbb{P}\left(Y_{t+1} \in \cdot \mid \mathcal{E}, X_t\right)$$

    3. update the buffer $\mathcal{D}_t \leftarrow \texttt{append}\left(\mathcal{D}_{t-1}, (X_t, Y_{t+1})\right)$
  **end for**
  compute the total regret incurred in $T$ time steps

$$\text{Regret}_j(T) = \sum_{t=1}^{T} \left(\overline{R}_* - \overline{R}_{X_t}\right)$$

**end for**
**return** average total regret $\frac{1}{J} \sum_{j=1}^{J} \text{Regret}_j(T)$

---

