# OpenReview forum: "Ensembles for Uncertainty Estimation: Benefits of Prior Functions and Bootstrapping"
_TMLR — Accepted by TMLR_

### Review · Reviewer_NvZp · 2022-10-28

**Summary Of Contributions:**

The authors investigate the influence of the use of random prior functions and bootstrapping on the performance of ensemble models. They show theoretically in the case of linear regression that ensemble models with prior functions and bootstrapping asymptotically converge to the true posterior, while ensembles without these do not necessarily and in fact remain bounded away from the true posterior under mild assumptions. They then show empirically that these benefits of prior functions and bootstrapping translate to performance gains on several real datasets, although not on all of them.

**Audience:**

Yes

**Broader Impact Concerns:**

I have no ethical concerns.

**Claims And Evidence:**

Yes

**Requested Changes:**

My main requested changes would be an improved discussion of the relevant related work and additional uncertainty evaluations on CIFAR10. Nice-to-have, but not 100% required, would be more realistic bandit experiments and one or two alternative diversity-inducing ensemble methods as baselines.

**Strengths And Weaknesses:**

Strengths:
- Ensembles are a popular model and this paper contributes to understanding them better
- The theoretical derivations seem to be correct
- The paper is well-written and the main ideas are clearly presented

Weaknesses:
- Some related works seem to be missing from the discussion
- The experiments could be improved towards more realistic problems

Major comments:
- Theoretical properties of ensembles with prior functions have previously been discussed in [1], so this should be mentioned
- There is a long line of research dealing with the topic of diversity in deep ensembles [e.g., 2-7], some of which also include certain theoretical convergence guarantees [e.g., 6]. Since the prior functions and bootstrapping studied in this paper also seem to mainly act by increasing diversity between the ensemble members, I think it should be discussed how this compares to these other proposed approaches to increase diversity. An experimental comparison would be great, but at least a theoretical discussion of the tradeoffs between these different methods seems warranted.
- I appreciate the experiments on CIFAR10, but since this paper deals with uncertainty estimation, it would be good to see some standard evaluations of the predictive uncertainties, such as uncertainty calibration on CIFAR10-C and OOD detection on SVHN/CIFAR-100, similar to, e.g., [8]. The current experiments are great and I appreciate the joint probability evaluation, but they are hard to compare with the existing literature.
- Given the theoretical results for linear regression, the bandit experiment is a nice confirmation of those, but the insights are not very realistic. It would be great to see a bandit experiment on a real problem, such as the ones in [9].

Minor comments:
- I feel like Props. 1 and 2 (or at least Prop. 1) could come a bit earlier in the text to improve the flow, since they are referenced one page before they appear.
- Why are there no error bars in Fig. 3?
- In Figs. 4 and 5, the y-axis is labeled "value", which could maybe rather be replaced by "nll" or similar and then the x-ticks could say "marginal" and "joint"
- The experiment environment is variously referred to as "neural testbed", "Neural Testbed", and "the Neural Testbed", which should be harmonized.
- Sec. 5: bit contrived -> a bit contrived
- Sec. 5: as loss -> as the loss
- Sec. 5.1: with same -> with the same
- Sec. 5.1: as generative -> as the generative

[1] https://openreview.net/forum?id=BJlahxHYDS

[2] https://arxiv.org/abs/1802.07881

[3] https://arxiv.org/abs/1912.02757

[4] https://ojs.aaai.org/index.php/AAAI/article/view/5849

[5] https://proceedings.neurips.cc/paper/2021/hash/466473650870501e3600d9a1b4ee5d44-Abstract.html

[6] https://proceedings.neurips.cc/paper/2021/hash/1c63926ebcabda26b5cdb31b5cc91efb-Abstract.html

[7] https://arxiv.org/abs/2101.05544

[8] https://arxiv.org/abs/2106.04015

[9] https://arxiv.org/abs/1802.09127

---

> ### Author Response · Authors · 2023-01-06
> **Author response**
>
> We thank you for your time and effort in this review process. Apologies for the delayed response. Thanks for pointing us to the relevant literature, we will add a detailed discussion in the paper. We address your specific questions below:
>
> > Theoretical properties of ensembles with prior functions have previously been discussed in [1], so this should be mentioned
>
> Thanks for pointing us to the relevant paper. We will discuss [1] in the revision of the paper. Though [1] does offer some theoretical insights, we believe that it has some major drawbacks, as detailed below:
> - First, [1] assumes that uncertainty is independent across different values of x. Ideally if we expect the uncertainty at x_1 is high, so should the uncertainty at x_2, if x_1 and x_2 are close to each other. However, this is not captured by the method proposed in [1].
> - Second, [1] assumes that uncertainty only depends on the input. Though this assumption is reasonable for the linear setting, it is quite restrictive for more complex models.
>
> > There is a long line of research dealing with the topic of diversity in deep ensembles [e.g., 2-7], some of which also include certain theoretical convergence guarantees [e.g., 6]... An experimental comparison would be great, but at least a theoretical discussion of the tradeoffs between these different methods seems warranted.
>
> Thanks for pointing us to the references. We have carefully reviewed them and will add a discussion on them in the paper revision. In particular, [2] proposes to train deep ensembles with a diversity function regularization. We have already discussed [3] in the paper. [4] proposes to train ensemble models for uncertainty estimation by maximizing “overall diversity”. [5] studies knowledge distillation (KD) for deep ensembles, and proposes a new approach based on Output Diversified Sampling (ODS). [6] studies repulsive deep ensembles, which added a kernelized repulsive term in the update rule of the deep ensembles. It also shows that the training dynamics of the proposed approach follow a Wasserstein gradient flow of the KL divergence to the true posterior. [7] introduces a novel training criterion called DICE, which increases the diversity by reducing spurious correlations among features.
>
> > I appreciate the experiments on CIFAR10, but since this paper deals with uncertainty estimation, it would be good to see some standard evaluations of the predictive uncertainties, such as uncertainty calibration on CIFAR10-C and OOD detection on SVHN/CIFAR-100, similar to, e.g., [8]. The current experiments are great and I appreciate the joint probability evaluation, but they are hard to compare with the existing literature.
>
> Thanks for the suggestion. We would like to point out that the low-data setting is similar in spirit to OOD. Our interest in joint predictive prediction is motivated by its relevance to downstream tasks (Wen et al., 2021: https://arxiv.org/pdf/2107.09224.pdf).
>
> > Given the theoretical results for linear regression, the bandit experiment is a nice confirmation of those, but the insights are not very realistic. It would be great to see a bandit experiment on a real problem, such as the ones in [9].
>
> We would like to clarify that this paper mainly focused on supervised learning problems, and the bandit experiment in the paper just shows that some key insights are carried over to the sequential decision problems. We agree that bandit experiments on real-world problems, such as those in [9], might deliver further insights. However, this is beyond the scope of this paper and we leave it to future work.
>
>
> > Minor comments
>
> We thank the reviewer for pointing these out. We will incorporate the suggested modifications.

---

> > ### Comment · Reviewer_NvZp · 2023-01-06
> > **Thanks**
> >
> > Thanks for the response and for adding the related work. I agree that the bandit experiments might be a bit of a stretch and can be left for future work. So once the results for the additional CIFAR experiments are updated in the manuscript, I'm happy to recommend acceptance.

---

> > > ### Author Response · Authors · 2023-01-08
> > > **Thanks**
> > >
> > > Thanks a lot for the prompt reply! We would like to clarify that the we have already included results on low data setting in the appendices. Please refer to Figure 9 for results on the neural testbed and Figure 10 for results on CIFAR. If you prefer, we are happy to include these results in the main body of the paper.

---

### Review · Reviewer_Vuf7 · 2022-11-04

**Summary Of Contributions:**

This paper discusses techniques to train ensemble models for uncertainty estimation in supervised and bandit learning.
Three existing techniques are considered: random initialization, random priors, and bootstrapping. The aim of the paper is to clarify the effectiveness of these methods when used alone or combined.
The two main claims of the paper are:

Claim 1: prior functions significantly improve joint predictions w.r.t. just using random initialization

Claim 2: bootstrapping is only useful (on top of the other techniques) when the input noise is significantly non-isotropic

The claims are substantiated both by theory and experiments. The theoretical results are in the special case of bayesian linear regression:
- Random initialization alone cannot represent any uncertainty, suggesting that random priors are necessary (Claim 1)
- Random priors + bootstrapping are sufficient to reconstruct the posterior distribution (Claim 1 and 2)
- When the input noise is significantly non-isotropic, random priors without bootstrapping do not allow to reconstruct the posterior distribution, suggesting that bootstrapping is necessary in general (Claim 2)

Experiments are conducted with ensembles of neural networks. In classification problems (Neural Testbed, CIFAR 10) marginal predictions are evaluated using the KL divergence and joint predictions are evaluated using the dyadic KL first proposed by Osband et al. (2022). Results show that indeed random priors improve predictions, especially when considering joint predictions and hyperparameter tuning is limited, and that bootstrapping further helps when the input is highly non-isotropic. The bandit experiment runs a version of Thompson sampling based on ensembles on an artificially generated task. Here significant improvement in the cumulated regret is shown by using random priors, and further improvement by using also bootstrapping.

**Audience:**

Yes

**Broader Impact Concerns:**

I don't think a broader impact statement is necessary for this paper.

**Claims And Evidence:**

Yes

**Requested Changes:**

Addressing weakness 1 is a critical adjustment. This can be done by considering the case of bootstrapping without random priors in the experiments or by discussing why this is a bad idea from a theoretical standpoint.

The following adjustments, instead, are just suggestions on how to improve the manuscript:
- To address weakness 2, you may test the different Thompson sampling variants on more realistic tasks like the ones considered in "A Contextual Bandit Bake-off" by Bietti et al. (2021), which are basically online-learning versions of classification problems and so should fit quite well within your framework
- I would elaborate more, in the introduction, on why uncertainty estimation is important and why ensemble methods are one of the most used approaches, and what are the pros and cons of other approaches
- I think evaluation metrics are a very important part of this work, and I suggest to give them more space in the main paper. For instance, I expect the concept of dyadic sampling to be new to most readers. Moving some of the content from appendix A to the main paper would be enough.
- The final expression for $\gamma_i$ on page 5 is confusing, since the argument of the conditional variance, $(X_t^\top v_i)(\theta_*^\top v_i)$, seems to be deterministic conditional on $X_t$
- Sentences that were unclear to me:  "the agent's imaginary environment" on page 3, "this might be bit contrived" on page 6

Finally, a question: is "uneven SNR" something that could be addressed with some sort of pre-processing of the data? Would this make bootstrapping useless (at least in the supervised learning setting)?

**Strengths And Weaknesses:**

Strenghts:
- The paper sheds some light on the role of different ensemble techniques, which, from the literature review by the authors, seems to be a matter of open discussion in the community. So, the results should be of interest to researchers and practitioners using ensemble methods in supervised or reinforcement learning.
- The claims are supported both by theory and experiments, and both appear to be sound and convincing.
- The paper is generally well written and clear.

Weaknesses:
1. The different techniques are considered in an incremental fashion: bootstrapping on top of random priors on top of random initialization. This is not fully motivated, and leaves some questions open, mostly what would be the effect of using bootstrapping without random priors. This case is considered neither in the theory nor in the experiments.
2. Only a randomly-generated linear bandit task is considered. Generating linear bandits in this way can lead to fundamentally easier problems than one would encounter in practice, as observed for instance in "Leveraging Good Representations in Linear Contextual Bandits" (Papini et al., 2021, Lemma 8). This is still an interesting and convincing illustrative example, but disappointingly limited when compared with the supervised learning experiments.

---

> ### Author Response · Authors · 2023-01-06
> **Author response**
>
> We thank the reviewer for their time and effort in the review process. Apologies for the delayed response. Responses to your questions:
>
>
>
> > The different techniques are considered in an incremental fashion: bootstrapping on top of random priors on top of random initialization. This is not fully motivated, and leaves some questions open, mostly what would be the effect of using bootstrapping without random priors. This case is considered neither in the theory nor in the experiments.
>
> The main reason why we didn’t consider bootstrapping alone is that bootstrapping cannot induce any uncertainty when there is little or no data. For example, consider an independent arm bandit problem/ cold-start problem, or a similar supervised learning problem, in this case, without a prior function, irrespective of the bootstrapping noise level, it cannot induce any uncertainty.  Thanks for this question, we will further clarify this in the paper.
>
> > Only a randomly-generated linear bandit task is considered. Generating linear bandits in this way can lead to fundamentally easier problems than one would encounter in practice, as observed for instance in "Leveraging Good Representations in Linear Contextual Bandits" (Papini et al., 2021, Lemma 8). This is still an interesting and convincing illustrative example, but disappointingly limited when compared with the supervised learning experiments.
>
> We appreciate that the reviewer thinks that our bandit example is interesting and convincing. Indeed, the experiment results on this simple bandit problem allude that even in such a simple problem, the consequences can be severe. We also would like to clarify that this paper mainly focused on supervised learning problems, and the bandit experiment in the paper just shows that some key insights are carried over to the sequential decision problems.
>
> > I would elaborate more, in the introduction, on why uncertainty estimation is important and why ensemble methods are one of the most used approaches, and what are the pros and cons of other approaches.
>
> Thanks, we will follow your advice and make necessary modifications.
>
> > I think evaluation metrics are a very important part of this work, and I suggest to give them more space in the main paper. For instance, I expect the concept of dyadic sampling to be new to most readers. Moving some of the content from appendix A to the main paper would be enough.
>
> Thanks, we will follow your advice and make necessary modifications.
>
> > The final expression for $\gamma_i$ on page 5 is confusing, since the argument of the conditional variance, $(X_t^\top\nu_i)(\theta_*^\top \nu_i)$, seems to be deterministic conditional on $X_t$
>
> In our setting, we consider $\theta_*$ to be unknown and a random variable, so even after conditioning on $X_t$, the quantity is still a random variable.
>
> > Sentences that were unclear to me: "the agent's imaginary environment" on page 3, "this might be bit contrived" on page 6
>
> We will modify them accordingly
>
> > Finally, a question: is "uneven SNR" something that could be addressed with some sort of pre-processing of the data? Would this make bootstrapping useless (at least in the supervised learning setting)?
>
> It is worth noting that most results in this paper are obtained under the assumption that the prior distribution is isometric (e.g. Footnote 2 discusses this assumption for the Bayesian linear regression problem). Though it is true that “uneven SNR” can be addressed by some sort of pre-processing of the data (e.g. normalizing the inputs), however, this will change the prior distribution and consequently violate the isometric prior assumption. Thus, in most cases, one cannot first pre-process the data to make the uneven SNR “even” and then use the results in this paper.
>
> In more general cases, we can show that bootstrapping is helpful if (1) the SNR is uneven and/or (2) the prior is “ill-conditioned”. Pre-processing of the data will just change an “uneven SNR” problem to a “ill-conditioned” prior problem. Consequently, pre-processing the data cannot make bootstrapping useless. We will further clarify this point in the revision.

---

### Review · Reviewer_DqPm · 2022-12-24

**Summary Of Contributions:**

The submission analyzes uncertainty estimation when using prior functions and bootstrapping with ensemble methods, aiming to show when these two ingredients improve over "vanilla" ensembles. They analyze a toy Bayesian linear regression example to show theoretical benefits of these two approaches, in particular benefits of bootstrapping in situations with different SNR across inputs.
They show improved empirical results in joint uncertainty estimation with small MLPs in the Neural Testbed setting as well as in image classification on CIFAR10, in particular illustrating the benefits of bootstrapping when artificially adding targeted label noise to simulate different SNRs across inputs.
Finally, they show improved regret when evaluating these techniques in a toy bandit problem (linear bandit with heteroskedatic noise) when using ensembles for Thompson sampling.

**Audience:**

Yes

**Claims And Evidence:**

No

**Requested Changes:**

See strengths/weakensses section for requested changes around more extensive empirical validation.

Minor Typos:

End of page 1: " In particular, accurate joint predictions are crucial for efficient exploration, adaptation and effective decision" effective decision -> effective decision making

End of section 1: "Then we justify the main points of this paper, as discussed above, via analaysis on Bayesian linear regression (Section 4)," analaysis -> analysis

2nd paragraph of sec 5: "This might be bit contrived as loss function is not convex in many deep learning problems."
be bit -> be a bit, add "the" before "loss function".

**Strengths And Weaknesses:**

While the paper attempts to answer questions that would be of interest to those working with uncertainty estimation in ML, I am ultimately not convinced that the claims of the paper (benefits of prior functions and bootstrapping) are adequately supported by the evidence. Theoretical analysis is limited to a toy linear regression problem, a setting where the baseline ensemble method "ensemble-N" trivially offers no uncertainty estimation benefits and is thus unlikely to be representative of machine learning/deep learning settings where such ensembling would be used. Empirical results consist of a setting with a small 2 layer MLP in NeuralTestbed, CIFAR10, and a linear bandit, which are all fairly toy tasks that are not necessarily meaningfully representative of realistic tasks in machine learning.

Specifically in comparisons between the baseline ensemble-N and prior functions (ensemble-P), I believe there is limited value in showing that prior functions leads to improved joint uncertainty esitmation in these toy tasks when the Osband 2018 [https://arxiv.org/pdf/1806.03335.pdf] which introduced randomized prior functions already demonstrate improved exploration over regular ensembles when applied in deep RL settings. I believe there would need to be a much more extensive empirical evaluation (larger scale supervised learning tasks, more complex reinforcement learning tasks than the low-dimensional linear bandit considered) for the results to be of interest. Additionally, it is interesting that in Fig 5 where non-uniform label noise is added, ensemble-P performs the same as ensemble-N in joint NLL (despite outperforming it in the clean dataset), a regression that should be investigated further.

For empirical results showing bootstrapping (ensemble-BP) improves when there is non-uniform SNR across inputs, my concern is that in all settings considered, there is explicit label noise being added to induce the non-uniform SNR. This synthetic non-uniformity may not be representative of realistic settings, and there should be a experiments validating this hypothesis in realistic tasks instead.

---

> ### Author Response · Authors · 2023-01-06
> **Author response**
>
> Thanks for your time and effort in reviewing our paper. We respond to your questions below.
>
> >While the paper attempts to answer questions that would be of interest to those working with uncertainty estimation in ML, I am ultimately not convinced that the claims of the paper (benefits of prior functions and bootstrapping) are adequately supported by the evidence. Theoretical analysis is limited to a toy linear regression problem, a setting where the baseline ensemble method "ensemble-N" trivially offers no uncertainty estimation benefits and is thus unlikely to be representative of machine learning/deep learning settings where such ensembling would be used. Empirical results consist of a setting with a small 2 layer MLP in NeuralTestbed, CIFAR10, and a linear bandit, which are all fairly toy tasks that are not necessarily meaningfully representative of realistic tasks in machine learning.
>
> We would like to clarify that along the research line of uncertainty modeling, CIFAR10 is used as a standard benchmark dataset and most of the earlier papers along this research line tried to justify their claims based on CIFAR10(ex: Nixon et al., 2020: https://openreview.net/pdf?id=dTCir0ceyv0, Lakshminarayan  et al., 2017: https://arxiv.org/pdf/1612.01474.pdf)
>
> > Specifically in comparisons between the baseline ensemble-N and prior functions (ensemble-P), I believe there is limited value in showing that prior functions leads to improved joint uncertainty esitmation in these toy tasks when the Osband 2018 [https://arxiv.org/pdf/1806.03335.pdf] which introduced randomized prior functions already demonstrate improved exploration over regular ensembles when applied in deep RL settings.
>
> Yes, as stated in the paper, our paper tried to address this debate on the advantages of prior functions and bootstrapping for ensembles. In spite of Osband 2018 demonstrating the advantages of prior functions, they have not been very popular in the deep learning community. We believe that this is due to, traditionally, metrics used for comparing the competency of the models are marginal metrics. With the use of metrics on joint predictions, which are more representative of downstream performance (https://arxiv.org/abs/2107.09224), we can clearly see the advantages of prior functions and bootstrapping.
>
> >  I believe there would need to be a much more extensive empirical evaluation (larger scale supervised learning tasks, more complex reinforcement learning tasks than the low-dimensional linear bandit considered) for the results to be of interest.
>
> Our understanding is that this paper is the first work to clarify the potential benefits of prior functions and bootstrapping, and the existing theoretical and experimental results suffice for this purpose. We would also like to keep the paper short to ensure the cognitive load to digest this paper is low. We agree that more experimental results might further support the points in the paper, however, they will make the paper longer and harder to digest, without adding any new insights.
>
> > Additionally, it is interesting that in Fig 5 where non-uniform label noise is added, ensemble-P performs the same as ensemble-N in joint NLL (despite outperforming it in the clean dataset), a regression that should be investigated further.
>
> Thanks for pointing this out. We suspect this is due to prior functions we are considering are randomly initialized networks. When such prior functions are used on classification problems with heteroskedastic noises (non-uniform label noises), it is not clear why they should lead to a significant performance improvement. With appropriate prior functions which reflect the class imbalance, we could see some performance improvement. We will try to clarify this in the paper.
>
> > For empirical results showing bootstrapping (ensemble-BP) improves when there is non-uniform SNR across inputs, my concern is that in all settings considered, there is explicit label noise being added to induce the non-uniform SNR. This synthetic non-uniformity may not be representative of realistic settings, and there should be experiments validating this hypothesis in realistic tasks instead.
>
> First, note that the main goal of this paper is to clarify the potential benefits of prior functions and bootstrapping. Though this synthetic example is simple, it suffices for this purpose. Second, there are many practical examples with non-uniform SNRs. For instance, for image classification problems, the correct labels for some images are very obvious (low noise), while the correct labels for other images can be very confusing (high noise).
>
> Thanks for pointing out the typos. We will rectify them.

---

### Decision · Action_Editors · 2023-03-20

**Recommendation:** Accept with minor revision

**Comment:**

The paper has received mixed reviews, with two reviewers leaning towards suggesting acceptance, and one reviewer arguing that the results of the paper are probably not interesting enough to warrant publication.

Reviewers Vuf7 and NvZp were more positive, although they have pointed out that the empirical analysis is not entirely satisfying. Both of them took issue with the limited bandit experiments, but after discussion they seem to be OK with this part being included in the paper (as long as the authors clearly indicate that they are intended as illustration of how the implications of this work extend to sequential decision making). Several other references were pointed out by Reviewer NvZp, and the authors promised to cover these in an updated version of the paper.

Reviewer DqPm was less positive about the paper, focusing their criticism on the relatively small scale of the experiments, and the Bayesian linear regression setting studied in the theoretical half of the paper. To an extent I see where this criticism comes from, but in my view, the results in this paper clearly show the advantage of bootstrapping for uncertainty estimation (as a comparison of the lower bound of Thm 1 with the positive result of Prop 1 reveals). Clearly, this is a lesson that holds more generally beyond the setting studied in the paper, the implication being that some existing uncertainty quantification methods may not be sufficiently good for estimating posterior distributions.

Based on my reading of the paper and the reviews, I think that this is a good paper that makes a clear point and illustrates it nicely with some experiments. Sure enough, the empirical results are not very large-scale but they do drive the point home cleanly, which would have been perhaps difficult to do with larger experiments.

Taking all of the above points into account, my conclusion is that the paper can be appropriate for publication at TMRL, given that the authors take the criticism raised in the reviews seriously, and actually implement the changes they promised in their responses.

**Audience:**

The topic of uncertainty quantification is timely and of great interest to the TMLR audience, and the results are new and interesting.

**Claims And Evidence:**

The paper provides clear evidence to all of its claims, in the form of mathematical proofs and carefully designed experiments.